# Precision-Recall Divergence Optimization for Generative Modeling with GANs and Normalizing Flows

**Alexandre Verine**
LAMSADE, CNRS,
Université Paris-Dauphine-PSL,
Paris, France
alexandre.verine@dauphine.psl.eu

**Benjamin Negrevergne**
LAMSADE, CNRS,
Université Paris-Dauphine-PSL,
Paris, France
benjamin.negrevergne@dauphine.psl.eu

**Muni Sreenivas Pydi**
LAMSADE, CNRS,
Université Paris-Dauphine-PSL,
Paris, France
muni.pydi@dauphine.psl.eu

**Yann Chevaleyre**
LAMSADE, CNRS,
Université Paris-Dauphine-PSL,
Paris, France
yann.chevaleyre@dauphine.psl.eu

## Abstract

Achieving a balance between image quality (precision) and diversity (recall) is a significant challenge in the domain of generative models. Current state-of-the-art models primarily rely on optimizing heuristics, such as the Fréchet Inception Distance. While recent developments have introduced principled methods for evaluating precision and recall, they have yet to be successfully integrated into the training of generative models. Our main contribution is a novel training method for generative models, such as Generative Adversarial Networks and Normalizing Flows, which explicitly optimizes a user-defined trade-off between precision and recall. More precisely, we show that achieving a specified precision-recall trade-off corresponds to minimizing a unique $f$-divergence from a family we call the *PR-divergences*. Conversely, any $f$-divergence can be written as a linear combination of PR-divergences and corresponds to a weighted precision-recall trade-off. Through comprehensive evaluations, we show that our approach improves the performance of existing state-of-the-art models like BigGAN in terms of either precision or recall when tested on datasets such as ImageNet.

## 1 Introduction

Evaluation of generative models has always been a challenging task. The metric used must reflect both the quality of the sample generated (precision) and how much the sample covers the targeted probability distribution (recall). Inception Score (IS) or Fréchet Inception Distance (FID) have been introduced and are now widely used by the community to select the best models. Typically, these metrics are favored because they *"correlate well with the visual fidelity of the samples"* and are *"sensitive to both the addition of spurious modes as well as mode dropping"* [44]. However as pointed by Kynkäänniemi et al. [28], *FID and IS group these two aspects into a single value without a clear trade-off.* Depending on the use-case, generative models might require a good precision (high-resolution image and video generation, artistic synthesis, 3D model design) or a good recall (data augmentation, drug discovery, anomaly detection). For that reason, a number of more principled methods [9, 15, 28, 44, 47] have emerged to assess precision and recall (hereafter abbreviated by

37th Conference on Neural Information Processing Systems (NeurIPS 2023).

P&R) independently, however these methods cannot be optimized during training, because they are not differentiable or because they are too computationally demanding.

To enhance either the precision or the recall of a particular model, an array of strategies and techniques have been employed. Usually these techniques involve altering the latent distribution *a posteriori* (e.g. truncation and temperature post-training). Other methods exists such as through rejection sampling, boosting, or instance selection [3, 12, 13, 19, 23, 33, 49]. Although these approaches may utilize proxies of P&R, they are not theoretically grounded in the principled methods of evaluating these metrics, leaving their alignment with the foundational concepts of P&R unverified. This leads to the following question.

**Question 1:** *Is it possible to train a generative model that achieves a specified trade-off between precision and recall?*

In addition to this question, another of our goals is to understand existing generative models in terms of P&R. Modern generative models, such as Generative Adversarial Networks (GANs) or Normalizing Flows (NFs), are typically designed to minimize specific divergence measures. Given a target distribution $P$ and a set of parameterized distributions $\{P_\theta | \theta \in \Theta\}$, a generative model aims to find the best fit $\widehat{P} = P_{\theta*}$ that minimizes a divergence between $P$ and $\widehat{P}$. These divergences induce different behaviors at convergence. For instance, optimizing the Kullback-Leibler (KL) divergence tends to favor mass-covering models [35], contrasting with the mode-seeking behavior observed with other generative models, leading to the infamous problem of mode collapse. Transitioning from one divergence to another does indeed alter the model's behavior in terms of precision and recall, however, the implicit trade-offs made during the optimization of a general divergence remain somewhat ambiguous. This motivates the following question.

**Question 2:** *What precision-recall trade-off does an arbitrary $f$-divergence minimize?*

In this paper, we bridge the gap between principled methods of P&R evaluation and controlling their trade-off. In doing so, we address Questions 1 and 2 by making the following contributions:

- We show that achieving a specified precision-recall trade-off corresponds to minimizing a particular $f$-divergence between $P$ and $\widehat{P}$. Specifically, in Theorem 4.3 we give a family of $f$-divergences (denoted by $\mathcal{D}_{\lambda\text{-PR}}$, $\lambda \in [0, \infty]$) that are associated with various points along the precision-recall curve of the generative model.

- We show that any arbitrary $f$-divergence can be written as a linear combination of $f$-divergences from the $\mathcal{D}_{\lambda\text{-PR}}$ family. This result makes explicit, the implicit precision-recall trade-offs made by generative models that minimize an arbitrary $f$-divergence.

- We propose a novel approach to train or fine-tune a generative model on notoriously hard to train $f$-divergences, with guarantees on divergences defined by the lipschitz constant of $f$.

- We use this approach to train models on a user specified trade-off between P&R by minimizing $\mathcal{D}_{\lambda\text{-PR}}$ for any given $\lambda$. We specifically focus on GANs and NFs. For instance, Figure 1 shows how our model performs under various settings of $\lambda$. With a high $\lambda$, we can train the model to favor precision over recall and vice-versa.

- Through extensive experiments, we show that our approach enables effective model training to minimize the PR-Divergence, particularly for fine-tuning pre-trained models, with a notable impact from the choice of trade-off parameters $\lambda$, while also demonstrating scalability with larger dimensions and datasets.

## 2 Related works

**Generative model evaluation metrics:**   IS or FID are widely adopted due to their ability to assess visual fidelity and sensitivity to mode variations. However, these metrics fall short in providing a trade-off between P&R. Principled approaches were introduced by Djolonga et al. [15] and Sajjadi et al. [44], and later extended by Simon et al. [47], providing a definition P&R independently using PR-Curves detailed in Section 3.3. We adopt this extended approach in our work. In image generative modeling, the current consensus for P&R evaluation are two methods. First, the method presented by Kynkäänniemi et al. [28], which provides a simpler evaluation of P&R based on the estimation

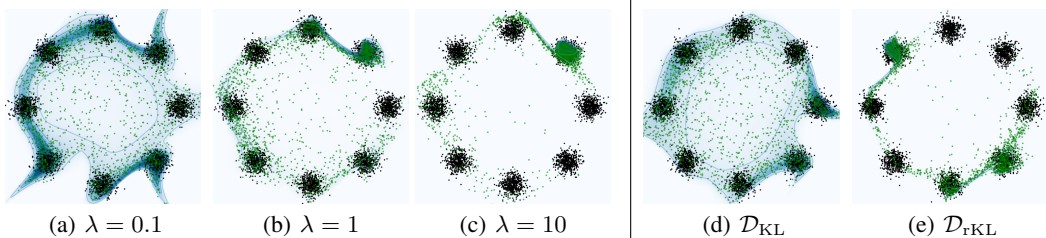

|     (a) $\lambda = 0.1$     |     (b) $\lambda = 1$     |     (c) $\lambda = 10$     |     (d) $\mathcal{D}_{\mathrm{KL}}$     |     (e) $\mathcal{D}_{\mathrm{rKL}}$     |

Figure 1: NFs - RealNVP [14] trained on 2D Gaussians. Fig. 1(a) to Fig. 1(c): models trained to minimize $\mathcal{D}_{\lambda\text{-PR}}$. Fig. 1(d) and Fig. 1(e): models trained to minimize $\mathcal{D}_{\mathrm{KL}}$ and $\mathcal{D}_{\mathrm{rKL}}$ respectively. Samples drawn from the true distribution $P$ are shown in black, samples drawn from the estimated distribution $\widehat{P}$ are shown in green and the log-likelihood of $\widehat{P}$ is shown in blue (darker means higher density). (a) $\lambda = 0.1$, favors recall over precision. (b) $\lambda = 1$, balanced precision vs. recall trade-off. (c) $\lambda = 10$, which favors precision over recall. As observed in [35], and demonstrated in Theorem 4.5, the $\mathcal{D}_{\mathrm{KL}}$ (1(d)) is mass covering and the $\mathcal{D}_{\mathrm{rKL}}$ (1(e)) in mode-seeking. The corresponding PR-Curve are presented in Figure 2(c).

of the support of the distribution using a k-NN algorithm, akin to the most recent work by Cheema and Urner [9]. Second, a method introduced by [36], computing the Density (D) and the Coverage (C) which account for local density and are robust to outliers in contrast to [28]. In natural language processing, one popular method, MAUVE [41], consists of the area under PR-Curves defined by [15]. However, because of their mathematical foundation, these methods remain unsuitable for training due to non-differentiability and high computational requirements.

**Trade-offs and techniques in generative model:** The challenge of balancing P&R in generative model training has led to a variety of techniques. Methods for enhancing precision often involve manipulating the latent distribution post-training, such as hard truncation [7, 24, 25, 46] or standard deviation adjustment, also called soft truncation [27]. One can also use rejection sampling in the image space [4, 50], in the latent space [23, 49], or in the dataset [13]. There are also numerous strategies to prevent mode collapse and boost recall [8, 31, 32], gradient-boosting methods [12, 16, 19] and mixture based latent models [6, 40]. However, it should be noted that these approaches lack a foundation in the principled methods of P&R evaluation. This underscores the need for a theoretically grounded approach to balance these crucial aspects in generative models, a gap that our current work aims to address. Note also that all these methods can be applied to any model, and in particular to the one trained using our proposed method.

**Divergence training in generative models:** Modern generative models, such as GANs and NFs, often employ specific divergence measures for model optimization. With works such as Grover et al. [20], Nowozin et al. [39], models can be trained with a variety of divergences such as $f$-divergence, thus observing different results of P&R. Another notable work by Midgley et al. [34] proposed training with $\alpha$-Divergence, under the assumption of access to data density. The work of [29], closely aligned with our approach, defines different orders of mode seeking and evaluates the corresponding $f$-divergences at these levels. Although its methodology employs the training of $f$-GAN models on simple datasets to illustrate the mode-seeking property, it does not offer the flexibility to establish a user-defined trade-off. The implicit trade-offs made by these divergence measures is still a challenge: the recent work of [48] shows the links between P&R and any DeGroot's divergences.

## 3  Background

**Notation:** Throughout the paper, we use $\mathcal{X} \subset \mathbb{R}^d$ to refer to the input space and $\mathcal{Z} \subset \mathbb{R}^m$ to the latent space of the model we consider. We also denote $\mathcal{P}(\mathcal{X})$ and $\mathcal{P}(\mathcal{Z})$ the set of all probability measures over measurable subsets of $\mathcal{X}$ and $\mathcal{Z}$ respectively. $P$ and $\widehat{P}$ are consistently used to denote the target and estimated distributions (both members of $\mathcal{P}(\mathcal{X})$). We also assume that $P$ and $\widehat{P}$ share the same support in $\mathcal{X}$, and that they admit densities denoted by the corresponding lower case letters $p$ and $\widehat{p}$, respectively. Finally, for any function $f \in \mathbb{R} \to \mathbb{R}$, we define the convex conjugate (or Fenchel transform) of $f$ given by $f^*(t) = \sup_{u \in \mathbb{R}} \{tu - f(u)\}$.

## 3.1 $f$-divergences

Given a convex lower semi-continuous (l.s.c) function $f : \mathbb{R}^+ \to \mathbb{R}$ satisfying $f(1) = 0$, the $f$-divergence between two probability distributions $P$ and $\widehat{P}$ is defined as follows.

$$\mathcal{D}_f(P\|\widehat{P}) = \int_{\mathcal{X}} \widehat{p}(\boldsymbol{x}) f\left(\frac{p(\boldsymbol{x})}{\widehat{p}(\boldsymbol{x})}\right) \mathrm{d}\boldsymbol{x}. \tag{1}$$

$\mathcal{D}_f$ is invariant to an affine transformation in $f$ i.e., $\mathcal{D}_f(P\|\widehat{P}) = D_{f^\dagger}(P\|\widehat{P})$ for $f^\dagger(u) = f(u) + c(u-1)$ for any constant $c \in \mathbb{R}$. Many well-known statistical divergences such as the KL divergence ($\mathcal{D}_{\mathrm{KL}}$), the reverse KL ($\mathcal{D}_{\mathrm{rKL}}$) or the Total Variation ($\mathcal{D}_{\mathrm{TV}}$) are $f$-divergences (see Table 1). Importantly, any $\mathcal{D}_f$ admits a dual variational form [37], with $\mathcal{T}$ denoting the set of all measurable functions $\mathcal{X} \to \mathbb{R}$:

$$\mathcal{D}_f(P\|\widehat{P}) = \sup_{T \in \mathcal{T}} \mathcal{D}_{f,T}^{\mathrm{dual}}(P\|\widehat{P}), \quad \text{where} \quad \mathcal{D}_{f,T}^{\mathrm{dual}}(P\|\widehat{P}) = \mathbb{E}_{\boldsymbol{x}\sim P}[T(\boldsymbol{x})] - \mathbb{E}_{\boldsymbol{x}\sim \widehat{P}}[f^*(T(\boldsymbol{x}))] \tag{2}$$

We use $T^{\mathrm{opt}} \in \mathcal{T}$ to denote the function that achieves the supremum in (15).

Table 1: List of common $f$-divergences. The generator $f$ is given with its Fenchel conjugate $f^*$. The optimal discriminator $T^{\mathrm{opt}}$ is given to compute the likelihood ratio $p(\boldsymbol{x})/\widehat{p}(\boldsymbol{x}) = \nabla f^*(T^{\mathrm{opt}}(\boldsymbol{x}))$. Then $f''(1/\lambda)/\lambda^3$ is given to compute the $\mathcal{D}_f$ as a combination of $\mathcal{D}_{\lambda\text{-PR}}$ using Theorem 4.4.

| DIVERGENCE | NOTATION | $f(u)$ | $f^*(t)$ | $T^{\mathrm{opt}}(\boldsymbol{x})$ | $f''(1/\lambda)/\lambda^3$ |
|---|---|---|---|---|---|
| KL | $\mathcal{D}_{\mathrm{KL}}$ | $u \log u$ | $\exp(t-1)$ | $1 + \log p(\boldsymbol{x})/\widehat{p}(\boldsymbol{x})$ | $1/\lambda^2$ |
| REVERSE KL | $\mathcal{D}_{\mathrm{rKL}}$ | $-\log u$ | $-1 - \log -t$ | $-\widehat{p}(\boldsymbol{x})p(\boldsymbol{x})$ | $1/\lambda$ |
| $\chi^2$-PEARSON | $\mathcal{D}_{\chi^2}$ | $(u-1)^2$ | $t^2/4 + t$ | $2\,(p(\boldsymbol{x})\widehat{p}(\boldsymbol{x}) - 1)$ | $2/\lambda^3$ |

## 3.2 Generative models

**Generative Adversarial Networks (GANs):** A GAN consists of two functions, generator $G : \mathcal{Z} \to \mathcal{X}$, discriminator $T : \mathcal{X} \to \mathbb{R}$, as well as a prior distribution $Q \in \mathcal{P}(\mathcal{Z})$ which is usually the standard normal $\mathcal{N}(0, \mathcal{I}_m)$. The estimated data distribution $\widehat{P}_G$ is the push-forward distribution of $Q$ by $G$. In the original work of Goodfellow et al. [17], a specific $\mathcal{D}_f$ is used to optimize the divergence between $P$ and $\widehat{P}_G$. In this paper, we consider the more general framework of Nowozin et al. [39] that can be used to train a GAN with any $\mathcal{D}_f$ by solving the following min-max optimization problem:

$$\min_G \max_T \mathcal{D}_{f,T}^{\mathrm{dual}}(P\|\widehat{P}_G) = \min_G \max_T \mathbb{E}_{\boldsymbol{x}\sim P}[T(\boldsymbol{x})] - \mathbb{E}_{\boldsymbol{x}\sim \widehat{P}_G}[f^*(T(\boldsymbol{x}))], \tag{3}$$

**Normalizing Flows (NFs):** An NF consists of an invertible function $G : \mathcal{Z} \to \mathcal{X}$ and a prior distribution $Q \in \mathcal{P}(\mathcal{Z})$. As for GANs, the estimated distribution $\widehat{P}_G$ is the push-forward of $Q$ by $G$. However, because $G$ is invertible, it is possible to compute the density $\hat{p}(\cdot)$ using a simple change of variable formula, $\hat{p}(\boldsymbol{x}) = q(G^{-1}(\boldsymbol{x}))|\det \mathrm{Jac}_{G^{-1}}(\boldsymbol{x})|$, where $\det \mathrm{Jac}_{G^{-1}}(\boldsymbol{x})$ is the determinant of the Jacobian matrix of $G^{-1}$ at $\boldsymbol{x}$, and train the generator using the $\mathcal{D}_{\mathrm{KL}}$ between $P$ and $\widehat{P}_F$, or equivalently, by maximizing the log-likelihood:

$$\min_G \mathcal{D}_{\mathrm{KL}}(P\|\widehat{P}_G) = H(P) - \max_G \mathbb{E}_{\boldsymbol{x}\sim P}[\log \hat{p}_G(\boldsymbol{x})], \tag{4}$$

where $H(P)$ is the continuous entropy of $P$. In practice, the generator function $G$ is typically represented by neural networks such as GLOW [27], RealNVP [42], or ResFlow [5, 10] for which $\det \mathrm{Jac}_{G^{-1}}(\boldsymbol{x})$ is easy to compute. Grover et al. [20] showed that it is possible to train an NF using most $f$-divergences. In practice, the log-likelihood is added to the min-max objective to stabilize learning. This gives us the following optimization problem:

$$\min_G \max_T \mathcal{D}_{f,T}^{\mathrm{dual}}(P\|\widehat{P}_G) - \gamma \mathbb{E}_{\boldsymbol{x}\sim P}[\log \hat{p}_G(\boldsymbol{x})], \tag{5}$$

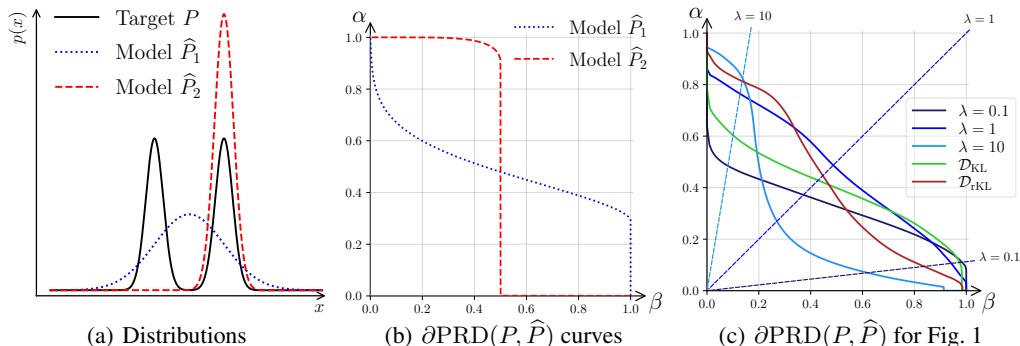

| (a) Distributions | (b) $\partial \mathrm{PRD}(P, \widehat{P})$ curves | (c) $\partial \mathrm{PRD}(P, \widehat{P})$ for Fig. 1 |

Figure 2: PR curves for two models $\widehat{P}_1$ and $\widehat{P}_2$ of $P$. Figure 2(a) shows $\widehat{P}_1, \widehat{P}_2$ and $P$. Figure 2(b) shows PR curves for $\widehat{P}_1, \widehat{P}_2$ against $P$. $\widehat{P}_1$ has good recall since it covers both modes of $P$, but low precision since it generates points between the modes. $\widehat{P}_2$ has good precision since it does not generate samples outside of $P$ but low recall since it can generate samples from only one mode. Figure 2(c) are the PR curves corresponding to Figure 1 and are detailed in Section 6.

### 3.3 Precision-Recall curve for generative models

Generative models are usually evaluated using a single criterion such as the IS [45] or the FID [21]. However, these criteria are unable to distinguish between the distinct failure modes of low precision (i.e., failure to produce quality samples) and low recall (i.e., failure to cover all modes of $P$). The following definition was introduced by Sajjadi et al. [44] and later extended by Simon et al. [47].

**Definition 3.1** (PRD set, adapted from Simon et al. [47]). *For $P, \widehat{P} \in \mathcal{P}(\mathcal{X})$, the P&R set $\mathrm{PRD}(P, \widehat{P})$ is defined as the set of pairs of precision $\alpha$ and recall $\beta$ in $\mathbb{R}^+ \times \mathbb{R}^+$ such that there exist $\mu \in \mathcal{P}(\mathcal{X})$ for which $P \geq \beta\mu$ and $\widehat{P} \geq \alpha\mu$. The* precision-recall curve *(or PR curve) is defined as $\partial \mathrm{PRD}(P, \widehat{P}) = \{(\alpha, \beta) \in \mathrm{PRD}(P, \widehat{P}) \mid \nexists(\alpha', \beta') \text{ with } \alpha' \geq \alpha \text{ and } \beta' \geq \beta\}$.*

An equivalent definition of $\mathrm{PRD}(P, \widehat{P})$ is found in Sajjadi et al. [44], where $(\alpha, \beta) \in \mathrm{PRD}(P, \widehat{P})$ if $P$ and $\widehat{P}$ can be decomposed as in (6) for some common component $\mu \in \mathcal{P}(\mathcal{X})$ and complementary components $\nu_P, \nu_{\widehat{P}} \in \mathcal{P}(\mathcal{X})$.

$$P = \alpha\mu + (1-\alpha)\nu_P \quad \text{and} \quad \widehat{P} = \beta\mu + (1-\beta)\nu_{\widehat{P}}. \tag{6}$$

Simon et al. [47] show that the PR curve is parameterized by $\lambda \in [0, +\infty]$ as $\partial \mathrm{PRD}(P, \widehat{P}) = \left\{ \alpha_\lambda(P\|\widehat{P}), \beta_\lambda(P\|\widehat{P}) \mid \lambda \in [0, +\infty] \right\}$, with $\alpha_\lambda(P\|\widehat{P}) = \int_{\mathcal{X}} \min\left(\lambda p(\boldsymbol{x}), \widehat{p}(\boldsymbol{x})\right) \mathrm{d}\boldsymbol{x}$ and $\beta_\lambda(P\|\widehat{P}) = \alpha_\lambda(P\|\widehat{P})/\lambda$. Here, $\lambda$ is called the *trade-off parameter* and can be used to adjust the sensitivity to precision or recall.

An illustration of the PR curve is given in Figure 2 for a target distribution $P$ that is a mixture of two Gaussians and two candidate models $\widehat{P}_1$ and $\widehat{P}_2$. We can see on Figure 2 that $\widehat{P}_1$ offers better results for large values of $\lambda$ (with high sensitivity to precision) whereas $\widehat{P}_2$ offers better results for low values of $\lambda$ (with high sensitivity to recall).

## 4 Precision and Recall trade-off as an $f$-divergence

In this section, we formalize the link between P&R trade-off and $f$-divergences, and address Question 2. We will exploit this link in Section 5 to train models that optimize a particular P&R trade-off.

### 4.1 Precision-Recall as an $f$-divergence

We start by introducing the *PR-Divergence* as follows.

**Definition 4.1** (PR-divergence). *Given a trade-off parameter $\lambda \in [0, +\infty]$, the PR-divergence (denoted by $\mathcal{D}_{\lambda\text{-PR}}$) is defined as $D_{f_\lambda}$ for $f_\lambda : \mathbb{R}^+ \to \mathbb{R}$ given by $f_\lambda(u) = \max(\lambda u, 1) - \max(\lambda, 1)$ for $\lambda \in \mathbb{R}^+$ and $f_\lambda(u) = 0$ for $\lambda = +\infty$.*

Note that $f_\lambda$ is continuous, convex, and satisfies $f_\lambda(1) = 0$ for all $\lambda$. A graphical representation of $f_\lambda$ can be found in Appendix B.2. The following proposition gives some properties of $\mathcal{D}_{\lambda\text{-PR}}$.

**Proposition 4.2** (Properties of the PR-Divergence).

- *The Fenchel conjugate $f_\lambda^*$ of $f_\lambda$ is defined on $\mathrm{dom}\,(f_\lambda^*) = [0, \lambda]$ and given by, $f_\lambda^*(t) = t/\lambda$ for $\lambda \leq 1$ and $f_\lambda^*(t) = t/\lambda + \lambda - 1$ otherwise.*

- *The optimal discriminator for the dual form is $T^{\mathrm{opt}}(\boldsymbol{x}) = \lambda \mathrm{sign}\{p(\boldsymbol{x})/\widehat{p}(\boldsymbol{x}) - 1\}$.*

- $\mathcal{D}_{\lambda\text{-PR}}(\widehat{P}\|P) = \lambda \mathcal{D}_{\frac{1}{\lambda}\text{-PR}}(P\|\widehat{P})$.

Observe that $\mathcal{D}_{\lambda\text{-PR}}(\widehat{P}\|P) \neq \mathcal{D}_{\lambda\text{-PR}}(P\|\widehat{P})$ in general, but for $\lambda = 1$, $\mathcal{D}_{1\text{-PR}}(P\|\widehat{P}) = \mathcal{D}_{1\text{-PR}}(\widehat{P}\|P) = \mathcal{D}_{\mathrm{TV}}(P\|\widehat{P})/2$. Having defined the PR-divergence, we can now show that P&R w.r.t $\lambda$ can be expressed as a function of the divergence between $P$ and $\widehat{P}$.

**Theorem 4.3** (P&R as a function of $\mathcal{D}_{\lambda\text{-PR}}$). *Given $P, \widehat{P} \in \mathcal{P}(\mathcal{X})$ and $\lambda \in [0, +\infty]$, the PR curve $\partial\mathrm{PRD}(P, \widehat{P})$ is related to the PR-divergence $\mathcal{D}_{\lambda\text{-PR}}(P\|\widehat{P})$ as follows.*

$$\alpha_\lambda(P\|\widehat{P}) = \min(1, \lambda) - \mathcal{D}_{\lambda\text{-PR}}(P\|\widehat{P}). \tag{7}$$

*Conversely, suppose that there exists a strictly decreasing linear function $h : [0,1] \to \mathbb{R}^+$ and an $f$-divergence $\mathcal{D}_f$ such that $h(\alpha_\lambda(P\|\widehat{P})) = \mathcal{D}_f(P\|\widehat{P})$ for all $P, \widehat{P} \in \mathcal{P}(\mathcal{X})$, then $f(u) = c_1 f_\lambda(u) + c_2(u-1)$.*

A direct consequence of Theorem 4.3 is that minimizing $\mathcal{D}_{\lambda\text{-PR}}$ is equivalent to maximizing $\alpha_\lambda$. In a more explicit way, Theorem 4.3 suggests that $\mathcal{D}_{\lambda\text{-PR}}$ is the *only* $f$-divergence (up to an affine transformation) for which this property holds. This makes $\mathcal{D}_{\lambda\text{-PR}}$ a uniquely suitable candidate for training a generative model with a specific P&R trade-off. The proof of Theorem 4.3 is in Appendix B.1

## 4.2 Relation between PR-divergences and other $f$-divergences

In the previous subsection, we showed that for each trade-off parameter $\lambda$, there exists a $\mathcal{D}_{\lambda\text{-PR}}$ that corresponds to optimizing for it. This raises the converse question of what trade-off is achieved by optimizing for an arbitrary $f$-divergence. We answer this by showing in the following theorem that any $f$-divergence can be expressed as a weighted sum of PR-divergences.

**Theorem 4.4** ($f$-divergence as weighted sums of PR-divergences). [1] *For any $P, \widehat{P} \in \mathcal{P}(\mathcal{X})$ supported on all of $\mathcal{X}$ and any $\lambda \in \mathbb{R}^+$, with $m = \min_{\mathcal{X}}(\widehat{p}(\boldsymbol{x})/p(\boldsymbol{x}))$ and $M = \max_{\mathcal{X}}(\widehat{p}(\boldsymbol{x})/p(\boldsymbol{x}))$:*

$$\mathcal{D}_f(P\|\widehat{P}) = \int_m^M \frac{1}{\lambda^3} f''\left(\frac{1}{\lambda}\right) \mathcal{D}_{\lambda\text{-PR}}(P\|\widehat{P})\mathrm{d}\lambda, \tag{8}$$

As a sanity check, observe that the weights $f''(1/\lambda)/\lambda^3$ remain invariant under an affine transformation in $f$ much like $D_f$.

**Corollary 4.5** ($\mathcal{D}_{\mathrm{KL}}$ and $\mathcal{D}_{\mathrm{rKL}}$ as an average of $\mathcal{D}_{\lambda\text{-PR}}$). *The $\mathcal{D}_{\mathrm{KL}}$ Divergence and the $\mathcal{D}_{\mathrm{rKL}}$ can be written as a weighted average of PR-Divergence $\mathcal{D}_{\lambda\text{-PR}}$ :*

$$\mathcal{D}_{\mathrm{KL}}(P\|\widehat{P}) = \int_m^M \frac{1}{\lambda^2} \mathcal{D}_{\lambda\text{-PR}}(P\|\widehat{P})\mathrm{d}\lambda, \quad and \quad \mathcal{D}_{\mathrm{rKL}}(P\|\widehat{P}) = \int_m^M \frac{1}{\lambda} \mathcal{D}_{\lambda\text{-PR}}(P\|\widehat{P})\mathrm{d}\lambda. \tag{9}$$

As we can see in this Corollary, both $\mathcal{D}_{\mathrm{KL}}$ and $\mathcal{D}_{\mathrm{rKL}}$ can be decomposed into a sum of PR-divergences terms $\mathcal{D}_{\lambda\text{-PR}}$, each weighted with $1/\lambda^2$ and $1/\lambda$ respectively. Note that if $\lambda < 1$ then $1/\lambda > 1/\lambda^2$, and conversely, if $\lambda > 1$. $\mathcal{D}_{\mathrm{KL}}$ is thus associating more weights on recall than $\mathcal{D}_{\mathrm{rKL}}$. This explains the *mass covering* behavior observed in NFs trained with $\mathcal{D}_{\mathrm{KL}}$. Comparatively, the $\mathcal{D}_{\mathrm{rKL}}$ assigns more weight to terms with a large lambda, leading to the *mode seeking* behavior empirically observed with flows trained with the $\mathcal{D}_{\mathrm{rKL}}$ [33]. However, as it will be observed in the Section 6, $\mathcal{D}_{\mathrm{rKL}}$ is still favoring low values of $\lambda$. Other weights for other $f$-divergences are in Appendix A.1.

---

[1] An equivalent result can be found in Corollary 19 of [48]

# 5 Minimizing the Precision-Recall divergence

In this section, we address Question 1 by introducing a new method that can be used to optimize a model with a specific precision-recall trade-off $\lambda$. A first naive strategy to achieve this is to use the $f$-GAN framework introduced by Nowozin et al. [39], and minimize the dual variational form of $\mathcal{D}_{\lambda\text{-PR}}$ presented in Theorem 4.3. Together, this results in solving the min-max problem:

$$\min_G \max_T \mathcal{D}_{f_\lambda, T}^{\text{dual}}(P\|\widehat{P}) = \min_G \max_T \mathbb{E}_{\boldsymbol{x} \sim P}\left[T(\boldsymbol{x})\right] - \mathbb{E}_{\boldsymbol{x} \sim \widehat{P}}\left[f_\lambda^*(T(\boldsymbol{x}))\right], \tag{10}$$

where $G$ and $T$ are both represented using neural networks. In practice, this strategy would fail because training a neural network with loss functions such as $f_{\text{TV}}^*$ (or in this case $f_\lambda^*$) is notoriously difficult due to vanishing gradients[2] that lead to poor training performance [20, 39]. Instead, training neural networks on functions $f^*$ such as $f_{\text{KL}}^*$ or $f_{\chi^2}^*$, results in much better empirical performance [29, 39, 51].

To avoid these issues, we show how to train the discriminator using an auxiliary divergence (based on a function $g \neq f$) to better estimate the target $f$-divergence. The main idea is to choose an auxiliary $g$-divergence that is adequate for training $T$ (e.g. $\mathcal{D}_{\text{KL}}$ or $\mathcal{D}_{\chi^2}$) and use it to compute the *likelihood ratio* $p(\boldsymbol{x})/\widehat{p}(\boldsymbol{x})$.

Because, at optimality, we have $\nabla g^*(T^{\text{opt}}(\boldsymbol{x})) = p(\boldsymbol{x})/\widehat{p}(\boldsymbol{x})$, we can then compute the $f$-divergence as follows:

$$\mathcal{D}_f(P\|\widehat{P}) = \int \widehat{p}(\boldsymbol{x}) f\left(\frac{p(\boldsymbol{x})}{\widehat{p}(\boldsymbol{x})}\right) \mathrm{d}\boldsymbol{x} = \int \widehat{p}(\boldsymbol{x}) f(\nabla g^*(T^{\text{opt}}(\boldsymbol{x}))) \mathrm{d}\boldsymbol{x}. \tag{11}$$

In practice, we do not have access to $T^{\text{opt}}$. Instead, we have a discriminator $T$ trained to maximize $\mathcal{D}_{g,T}^{\text{dual}}$. At any time during training, we can estimate the $f$-divergence based on $T$ using the *primal estimate*, which we define as follows.

**Definition 5.1** (Primal estimate $\mathcal{D}_{f,T}^{\text{primal}}$). *Let $P, \widehat{P} \in \mathcal{P}(\mathcal{X})$. For any function $T : \mathcal{X} \to \mathbb{R}$, $f : \mathbb{R}^+ \to \mathbb{R}$, and $g : \mathbb{R}^+ \to \mathbb{R}$, we define the primal estimate $\mathcal{D}_{f,T}^{\text{primal}}$ as follows.*

$$\mathcal{D}_{f,T}^{\text{primal}}(P\|\widehat{P}) = \int_\mathcal{X} \widehat{p}(\boldsymbol{x}) f\left(r(\boldsymbol{x})\right) \mathrm{d}\boldsymbol{x}, \tag{12}$$

*where $r : \mathcal{X} \to \mathbb{R}^+$ is given by, $r(\boldsymbol{x}) = \nabla g^*(T(\boldsymbol{x}))$.*

The success of this approach depends on how well $r$ approximates $p(x)/\widehat{p}(x)$, which depends on $T$. We first show that the approximation error of $r$ measured in terms of the Bregman divergence [2] (also defined in Appendix A.2). It corresponds *exactly* to the approximation error of $\mathcal{D}_{g,T}^{\text{dual}}$, so minimizing the latter will also minimize the former.

**Theorem 5.2** (Error of the estimation of an $f$-divergence under the dual form.). *For any discriminator $T : \mathcal{X} \to \mathbb{R}$ and $r(\boldsymbol{x}) = \nabla f^*(T(\boldsymbol{x}))$,*

$$\mathcal{D}_g(P\|\widehat{P}) - \mathcal{D}_{g,T}^{\text{dual}}(P\|\widehat{P}) = \mathbb{E}_{\widehat{P}}\left[\text{Breg}_g\left(r(\boldsymbol{x}), \frac{p(\boldsymbol{x})}{\widehat{p}(\boldsymbol{x})}\right)\right]. \tag{13}$$

On the basis of this result, the quality of the approximation will crucially depend on $\nabla g$. We can show that if the auxiliary $g$ is strongly convex, the error in the estimation of $\mathcal{D}_{f,T}^{\text{primal}}$ is bounded:

**Theorem 5.3** (Bound on the estimation of an $f$-divergence using an auxiliary $g$-divergence). *Let $f, g : \mathbb{R}^+ \to \mathbb{R}$ be such that $g$ is $\mu$-strongly convex and $f$ is $\sigma$-Lipschitz. For the discriminator $T : \mathcal{X} \to \mathbb{R}$, let $r(\boldsymbol{x}) = \nabla g^*(T(\boldsymbol{x}))$. Then*

$$\mathcal{D}_g(P\|\widehat{P}) - \mathcal{D}_{g,T}^{\text{dual}} \leq \epsilon \implies \left|\mathcal{D}_f(P\|\widehat{P}) - \mathcal{D}_{f,T}^{\text{primal}}(P\|\widehat{P})\right| \leq \sigma\sqrt{\frac{2\epsilon}{\mu}}.$$

---

[2]The explanation for this vanishing gradient phenomenon primarily relies on the fact that the optimal discriminator for these functions is $T^{\text{opt}}(\boldsymbol{x}) = \text{sign}(\widehat{p}(\boldsymbol{x})/p(\boldsymbol{x}) - 1)$. To approximate it, [39] recommends the use of a *tanh* activation function that is known to induce vanishing gradients.

If $T$ successfully maximizes $\mathcal{D}_{g,T}^{\text{dual}}$, then the primal estimation converges to $\mathcal{D}_f$. To implement this approach, we propose the following simplified version of the algorithm: Repeat until convergence these 3 steps:

1. Let $x_1^{\text{real}} \dots x_N^{\text{real}} \sim P$ and $x_1^{\text{fake}} \dots x_N^{\text{fake}} \sim \widehat{P}_G$.
2. Update the parameters of $T$ by ascending the gradient

$$\nabla \mathcal{L}_T = \frac{1}{N} \nabla \left\{ \sum_{i=1}^N T\left(x_i^{\text{real}}\right) - \sum_{i=1}^N g^*\left(T(x_i^{\text{fake}})\right) \right\}.$$

3. Update the parameters of $G$ by descending the gradient

$$\nabla \mathcal{L}_G = \frac{1}{N} \nabla \left\{ \sum_{i=1}^N f\left(\nabla g^*\left(T(x_i^{\text{fake}})\right)\right) \right\}.$$

This method closely parallels the GAN training procedure, with the key distinction that T and G optimize objectives based on different $f$. In practice, our implementation uses a stochastic gradient descent, fully detailed in Algorithm 1 in Appendix C.

## 6 Experiments

In this section, we employ the auxiliary loss approach outlined in Section 5 to train various models.

Specifically, we train NFs on 2D synthetic data, MNIST and FashionMNIST, while we train BigGAN on CIFAR-10, CelebA64, ImageNet128, and FFHQ256. All models are tested in terms of IS and FID with 50k samples, and on P&R with 10k samples using the method of Kynkäänniemi et al. [28] with $k = 3$ for MNIST and FashionMNIST and with $k = 5$ for CelebA64, ImageNet128 and FFHQ256. Also, we test every model in terms of Density and Coverage [36] on 10k samples with $k = 5$. In this paper, we present a selection of experimental results. For a comprehensive set of results, including model parameters, optimizers, learning rates, and samples, please refer to Appendix D. We also included in this Appendix a set of experiments run using the naive approach based on Equation 10, thus showing that the discriminator fails to train as explained in Section 5. To ensure reproducibility, our models and code are available on the GitHub repository of the project[3].

We show that: 1) the auxiliary loss approach effectively enables the training of a model to minimize the PR-Divergence, 2) this method is suitable for fine-tuning pre-trained models, 3) the choice of trade-off parameter $\lambda$ significantly influences the results on P&R, and finally, 4) our method scales well with larger dimensions and datasets.

**Normalizing Flows on synthetic data:** NFs are typically trained to minimize $\mathcal{D}_{\text{KL}}$, in addition to their structural limitation [11, 52], resulting in good recall but poor precision. Prior work has employed various techniques [27, 49] to improve model precision post-training, we use our method to directly train the model on a given trade-off. We demonstrate our approach by training RealNVP models on a 2D synthetic dataset using $\mathcal{D}_{\lambda\text{-PR}}$ with various $\lambda$ values and using $\mathcal{D}_{\text{KL}}$ and $\mathcal{D}_{\text{rKL}}$ for baseline comparison. As we can see in Figure 1, increasing $\lambda$ leads to an increase in precision in the resulting models. Using our $\mathcal{D}_{\lambda\text{-PR}}$ estimation, we compute the corresponding PR curves (Figure 2(c)). The $\lambda = 0.1$ model, while best at $\alpha_{0.1}$, performs poorly at $\alpha_1$ and $\alpha_{10}$, clearly demonstrating the impact of maximizing $\alpha_\lambda$. This pattern across models validates the efficacy of our method in minimizing the desired trade-off.

**Using $\mathcal{D}_{\text{KL}}$ vs $\mathcal{D}_{\chi^2}$ on MNIST and FashionMNIST:** We now demonstrate that we can improve the precision by directly minimizing the $\mathcal{D}_{\lambda\text{-PR}}$ with the correct $\lambda$ using pre-trained GLOW models [27] on both MNIST [54] and FashionMNIST [53]. Figure 3 and Figure 4 present the samples obtained. First, we observe that while $\lambda$ increases, the visual quality improves, but the models focus on a few modes only (0, 1, 7, 6, 9 for MNIST and "Trouser" for FashionMNIST). Then training with both $\mathcal{D}_{\text{KL}}$ and $\mathcal{D}_{\text{rKL}}$ divergences aligns with our expectations: $\mathcal{D}_{\text{KL}}$ training leads to high recall, while $\mathcal{D}_{\text{rKL}}$, to higher precision. However, according to Corollary 4.5, $\mathcal{D}_{\text{rKL}}$ still favors low $\lambda$ values; consequently, our models trained with $\lambda > 0.1$ demonstrate better precision than standard flow-GAN models. Furthermore, we find that both auxiliary $g$ functions ($f_{\text{KL}}$ and $f_{\chi^2}$) used to train the discriminator $T$ perform well. In practice, training with $f_{\chi^2}$ proves to be more stable, particularly for FashionMNIST, with results reported in Appendix D.3. For larger models, we use exclusively $g = f_{\chi^2}$.

---

[3]`https://github.com/AlexVerine/PrecisionRecallBigGan`

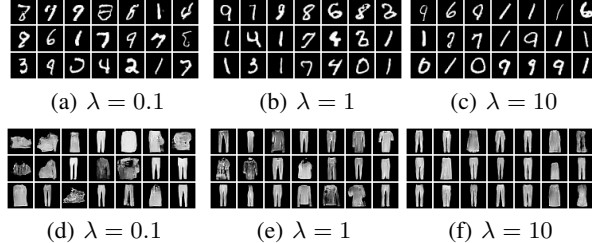

(a) $\lambda = 0.1$     (b) $\lambda = 1$     (c) $\lambda = 10$

(d) $\lambda = 0.1$     (e) $\lambda = 1$     (f) $\lambda = 10$

Figure 3: Samples from NFs - GLOW trained on MNIST (3(a) to 3(c)) and FashionMNIST (3(d) to 3(f)). Recall decreases as precision increases for $\lambda$ between 0.1 and 10.

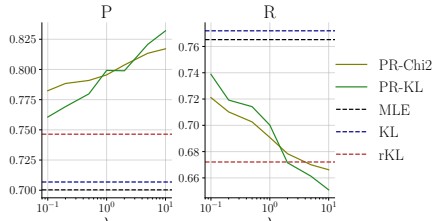

Figure 4: Precision and Recall as a function on $\lambda$ for GLOW trained on MNIST with $T$ trained with $f_{KL}$ or $f_{\chi^2}$, and for models trained $\mathcal{D}_{KL}$, $\mathcal{D}_{rKL}$ and MLE.

**Training BigGAN on CIFAR-10 and CelebA64** Now we demonstrate that our method can also be used to train large generative models. Our choice to adopt the BigGAN architecture [7] was informed by several factors: its competitive performance close to state-of-the-art models; its versatility, permitting diverse experimental explorations; and the fact that it is publicly accessible, ensuring experiment reproducibility. We train Big-GAN using both the baseline method (i.e., hinge loss) and our proposed method on CIFAR-10 [1] and CelebA64 [30]. A notable observation when training with different precision-recall trade-offs is the early elimination of modes from the target distribution at higher values of $\lambda$. As illustrated in Figure 5, models with low values of $\lambda$ converge to achieve maximum recall, while those with $\lambda > 1$ rapidly saturate to a lower recall value. A similar behavior can be observed for models trained on CelebA, as shown in Figure D.18. In Table 2, we present the quantitative metrics (Precision, Recall, and FID) for the baseline BigGAN, the BigGAN models trained with varying trade-offs and the current state-of-the-art models: EDM-G++ [26] for CIFAR-10 and ADM-IP [38] for CelebA64. Employing our proposed method enables us to adjust the trade-off, allowing us to train models that closely approach the state-of-the-art recall and, for high $\lambda$, even outperform state-of-the-art models in terms of precision.

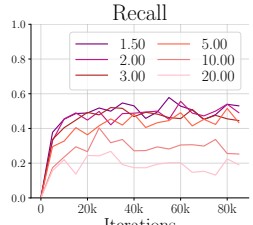

Figure 5: Recall during training of BigGAN optimizing different trade-off on CIFAR-10. Models for high $\lambda$ saturates in an early stage at decreasing values of recall.

Table 2: BigGAN trained with the vanilla approach [7] and with a variety of $\lambda$ using our approach on CIFAR-10 and CelebA64. We compare our approach with hard truncation on the baseline model. FID ($\downarrow$), Precision ($\uparrow$), Recall ($\uparrow$), Density ($\uparrow$) and Coverage ($\uparrow$) are reported. In **bold**, our best model is highlighted and the state-of-the-art FID is marked with an exponent $^*$.

| MODEL | CIFAR-10 $32 \times 32$ | | | | | CELEBA $64 \times 64$ | | | | |
|---|---|---|---|---|---|---|---|---|---|---|
| | FID | P | R | D | C | FID | P | R | D | C |
| BASELINE BIGGAN | 13.37 | 86.51 | 65.66 | 0.76 | 0.81 | 9.16 | 78.41 | **51.42** | 0.89 | 0.48 |
| HARD $\psi = 2.0$ | 13.95 | 86.82 | 63.58 | 0.77 | 0.79 | 10.60 | 80.81 | 48.21 | 0.96 | 0.50 |
| HARD $\psi = 1.0$ | 17.23 | 88.03 | 53.63 | 0.83 | 0.75 | 17.97 | **84.30** | 37.46 | 1.11 | 0.49 |
| HARD $\psi = 0.5$ | 20.11 | 87.87 | 44.98 | 0.83 | 0.70 | 25.70 | 83.70 | 28.81 | 1.08 | 0.42 |
| $\lambda = 0.05$ | 13.29 | 81.10 | 70.63 | 0.61 | 0.80 | - | - | - | - | - |
| $\lambda = 0.1$ | **11.62** | 81.78 | **74.58** | 0.66 | **0.83** | - | - | - | - | - |
| $\lambda = 0.2$ | 13.36 | 84.85 | 65.13 | 0.74 | 0.82 | 8.79 | 83.37 | 44.07 | 1.09 | **0.54** |
| $\lambda = 0.5$ | 14.50 | 83.27 | 68.23 | 0.70 | 0.81 | **6.03** | 77.60 | 55.98 | 0.88 | 0.50 |
| $\lambda = 1.0$ | 14.03 | 83.04 | 69.35 | 0.68 | 0.79 | 13.07 | 81.70 | 36.85 | 1.00 | 0.47 |
| $\lambda = 2.0$ | 16.94 | 84.93 | 59.79 | 0.75 | 0.78 | 14.23 | 82.98 | 32.87 | 1.16 | 0.49 |
| $\lambda = 5.0$ | 32.54 | 83.39 | 56.94 | 0.68 | 0.73 | 22.45 | 83.96 | 25.81 | **1.21** | 0.43 |
| $\lambda = 10.0$ | 39.69 | 84.11 | 39.29 | 0.75 | 0.67 | - | - | - | - | - |
| $\lambda = 20.0$ | 67.03 | **90.03** | 21.81 | **0.98** | 0.56 | - | - | - | - | - |
| DENSEFLOW [18] | — | 88.90 | 60.81 | 0.86 | 0.71 | — | 85.83 | 38.22 | 1.17 | 0.82 |
| ADM-IP [38] | 3.25 | 80.67 | 83.65 | 0.65 | 0.87 | $1.53^*$ | 23.42 | 64.48 | 0.09 | 0.24 |
| EDM G++ [26] | $1.77^*$ | 78.48 | 85.83 | 0.60 | 0.87 | - | - | - | - | - |
| STYLEGAN-XL [46] | 1.85 | 85.11 | 70.04 | 0.75 | 0.85 | - | - | - | - | - |

In every experiment, we compare our approach with traditional post-training techniques. In Table 2, we give the results for the hard truncation (also called *the truncation trick* in [7]) of the latent distribution $Q$ for different $\psi$. We observe that this method enables to improve solely the precision by trading off the recall; however, note that the truncation can be use in addition to our approach.

**Fine-tuning BigGAN on Imagenet128 and FFHQ**   Finally, we apply our method to pre-trained BigGAN models. To accomplish this, we implement a straightforward technique: initially, we train the discriminator for a brief period, allowing the model to transition from the vanilla training objective to the $\mathcal{D}_g^{\text{dual}}$. This approach enables us to train BigGAN on large datasets such as ImageNet128 [43] and datasets with high dimensions such as FFHQ256 [24]. We compare our method with both hard truncation and soft truncation (also denoted temperature in [27]). Both methods can be used in addition to our approach. The metrics presented in Table 3 demonstrate that (1) we enhance a given model's precision (by $+2.83\%$) or recall (by $+1.17\%$) on ImageNet, thereby achieving state-of-the-art precision, and (2) our method compromises less on the trade-off than truncation. For instance, in FFHQ, for a similar precision improvement ($\approx +15.5\%$), recall is decreased by more than $5\%$ for truncation methods and only by $1.65\%$ with our approach.

Table 3: BigGAN fine-tune with the vanilla approach [7] and with a variety of $\lambda$ using our approach on ImageNet128 and FFHQ256. We compare our approach with hard truncation on the baseline model. FID ($\downarrow$), Precision ($\uparrow$), Recall ($\uparrow$), Density ($\uparrow$) and Coverage ($\uparrow$) are reported. In **bold**, our best model is highlighted and the state-of-the-art FID is marked with an exponent $^*$.

| MODEL | IMAGENET $128 \times 128$ | | | | | FFHQ $256 \times 256$ | | | | |
|---|---|---|---|---|---|---|---|---|---|---|
| | FID | P | R | D | C | FID | P | R | D | C |
| BASELINE BIGGAN | 9.83 | 28.04 | 41.21 | 0.14 | 0.17 | 41.41 | 65.57 | **10.17** | 0.52 | 0.47 |
| SOFT $\psi = 0.7$ | 11.39 | 23.04 | 31.13 | 0.11 | 0.15 | 56.43 | 76.59 | 4.87 | 0.70 | 0.41 |
| SOFT $\psi = 0.5$ | 15.49 | 20.20 | 19.83 | 0.10 | 0.14 | 82.05 | 84.48 | 1.58 | 0.89 | 0.32 |
| HARD $\psi = 2.0$ | **9.69** | 25.83 | 39.89 | 0.13 | **0.18** | 43.32 | 68.84 | 8.66 | 0.58 | 0.47 |
| HARD $\psi = 1.0$ | 12.12 | 21.86 | 35.42 | 0.11 | 0.15 | 56.19 | 76.44 | 4.76 | 0.75 | 0.44 |
| HARD $\psi = 0.5$ | 15.21 | 21.13 | 29.55 | 0.10 | 0.13 | 71.32 | 80.99 | 4.84 | 0.84 | 0.36 |
| $\lambda = 0.2$ | 9.92 | 26.69 | 42.04 | 0.13 | 0.17 | 35.66 | 78.70 | 9.45 | 0.88 | 0.60 |
| $\lambda = 0.5$ | 10.82 | 26.83 | **42.38** | 0.13 | 0.16 | 35.24 | 78.41 | 9.66 | 0.89 | 0.60 |
| $\lambda = 1.0$ | 20.42 | 29.72 | 28.21 | **0.15** | 0.15 | 35.91 | 78.95 | 8.32 | 0.90 | 0.57 |
| $\lambda = 2.0$ | 20.21 | 30.27 | 30.49 | 0.14 | 0.14 | 36.33 | 81.10 | 8.69 | 1.05 | **0.64** |
| $\lambda = 5.0$ | 20.76 | **30.87** | 28.38 | **0.15** | 0.15 | 38.16 | **84.31** | 8.52 | **1.15** | 0.63 |
| ADM [22] | 2.97 | 26.63 | 68.54 | 0.14 | 0.16 | - | - | - | - | - |
| STYLEGAN-XL [46] | 1.81$^*$ | 11.35 | 68.04 | 0.04 | 0.09 | 2.19$^*$ | 79.91 | 38.79 | 0.86 | 0.73 |

## 7   Conclusion

In this paper, we present a novel method for training generative models using a new PR-divergence, $\mathcal{D}_{\lambda\text{-PR}}$. Our approach offers a unique advantage over existing methods as it allows for explicit control of the precision-recall trade-off in generative models. By varying the trade-off parameter $\lambda$, one can train a variety of models ranging from mode seeking (high precision) to mode covering (high recall), as well as more balanced models that may be more suitable for various applications. Through extensive experiments, we demonstrate the validity of our method and show that it scales well with larger dimensions and datasets. Our approach also provides insights into the implicit P&R trade-offs made by models trained with other $f$-divergences. By introducing the $\mathcal{D}_{\lambda\text{-PR}}$ divergence and providing a systematic approach for training generative models based on user-specified trade-offs, we contribute to the development of more customizable generative models. Our method currently applies to GANs and NFs only, it is still unclear if a similar approach can be applied to trained diffusion models: a promising work [26] uses a discriminator to refine diffusion models, and could be used to estimate the $\mathcal{D}_{\lambda\text{-PR}}$.

## Acknowledgment

We are grateful for the grant of access to computing resources at the IDRIS Jean Zay cluster under allocations No. AD011011296 and No. AD011014053 made by GENCI.

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

# A Supplementary background

## A.1 $f$-divergences

$f$-divergence between two probability distributions $P$ and $Q$ over a common support $\mathcal{X}$ is defined as:

$$\mathcal{D}_f(P\|Q) = \int_{\mathcal{X}} q(\boldsymbol{x}) f\left(\frac{p(\boldsymbol{x})}{q(\boldsymbol{x})}\right) \, \mathrm{d}\boldsymbol{x}, \tag{14}$$

where $p(\boldsymbol{x})$ and $q(\boldsymbol{x})$ denote the densities of $P$ and $Q$, respectively, and $f$ is a convex function such that $f(1) = 0$. Any $\mathcal{D}_f$ admits a dual variational form [37], with $\mathcal{T}$ denoting the set of all measurable functions $\mathcal{X} \to \mathbb{R}$:

$$\mathcal{D}_f(P\|\widehat{P}) = \sup_{T \in \mathcal{T}} \left( \mathbb{E}_P\left[T(\boldsymbol{x})\right] - \mathbb{E}_{\widehat{P}}\left[f^*(T(\boldsymbol{x}))\right] \right), \tag{15}$$

The properties of $f$-divergence are as follows:

- **Non-Negativity:** For any two probability distributions $P$ and $\widehat{P}$ on $\mathcal{X}$, we have $\mathcal{D}_f(P\|\widehat{P}) \geq 0$. The equality holds if and only if $P = \widehat{P}$.

- **Convexity in $f$:** If $f(u)$ is a convex function, then $\mathcal{D}_f(P\|\widehat{P})$ is convex in the input distributions $P$ and $\widehat{P}$.

- **Non-Symmetry:** Generally, $f$-divergence is not symmetric, i.e., $\mathcal{D}_f(P\|\widehat{P}) \neq \mathcal{D}_f(\widehat{P}\|P)$.

- **Linearity:** $f$-divergence is linear in $f$, i.e., for any two convex functions $f_1(u)$ and $f_2(u)$, and for any two real numbers $a$ and $b$, the divergence $\mathcal{D}_{af_1+bf_2}(P\|\widehat{P}) = a\mathcal{D}_{f_1}(P\|\widehat{P}) + b\mathcal{D}_{f_2}(P\|\widehat{P})$.

Specific choices of the function $f$ in the $f$-divergence definition can yield various well-known divergence measures. We will review some of them of the principal properties. For every divergence, we report here the generator function $f$, its convex conjugate $f^*$, the domain of the convex conjugate $\mathrm{dom}(f^*)$, the typical activation function used to ensure that $T(\boldsymbol{x}) \in \mathrm{dom}(f^*)$ and finally the optimal discriminator $T^*$.

**Kullback Leibler:**

- Notation: $\mathcal{D}_{\mathrm{KL}}$,
- Generator Function: $f(u) = u\log(u)$,
- Convex Conjugate domain: $\mathrm{dom}(f^*) = \mathbb{R}$,
- Convex Conjugate: $f^*(t) = \exp(t-1)$,
- Activation: $v \mapsto v$,
- Optimal discriminator: $T^*(\boldsymbol{x}) = 1 + \log\left(\frac{p(\boldsymbol{x})}{\widehat{p}(\boldsymbol{x})}\right)$.

**Reverse Kullback Leibler:**

- Notation: $\mathcal{D}_{\mathrm{rKL}}$,
- Generator Function: $f(u) = -\log(u)$,
- Convex Conjugate domain: $\mathrm{dom}(f^*) = \mathbb{R}^-$,
- Convex Conjugate: $f^*(t) = -1 - \log -t$,
- Activation: $v \mapsto -\exp v$,
- Optimal discriminator: $T^*(\boldsymbol{x}) = -\frac{\widehat{p}(\boldsymbol{x})}{p(\boldsymbol{x})}$.

$\chi^2$ **Pearson:**

- Notation: $\mathcal{D}_{\chi^2}$,
- Generator Function: $f(u) = (u-1)^2$,
- Convex Conjugate domain: $\mathrm{dom}(f^*) = \mathbb{R}$,
- Convex Conjugate: $f^*(t) = t^2/4 + t$,
- Activation: $v \mapsto v$,
- Optimal discriminator: $T^*(\boldsymbol{x}) = 2\left(\frac{p(\boldsymbol{x})}{\widehat{p}(\boldsymbol{x})} - 1\right)$.

**GAN:**

- Notation: $\mathcal{D}_{\mathrm{GAN}}$,
- Generator Function: $f(u) = u\log(u) - (u+1)\log(u+1)$,
- Convex Conjugate domain: $\mathrm{dom}(f^*) = \mathbb{R}^-$,
- Convex Conjugate: $f^*(t) = -\log(1 - \exp(t))$,
- Activation: $v \mapsto -\exp - \log(1 - \exp(v))$,
- Optimal discriminator: $T^*(\boldsymbol{x}) = \log\frac{p(\boldsymbol{x})}{p(\boldsymbol{x}) + \widehat{p}(\boldsymbol{x})}$.

**Total Variation:**

- Notation: $\mathcal{D}_{\mathrm{TV}}$,
- Generator Function: $f(u) = |u - 1|$,
- Convex Conjugate domain: $\mathrm{dom}(f^*) = \left[-\frac{1}{2}, \frac{1}{2}\right]$,
- Convex Conjugate: $f^*(t) = t$,
- Activation: $v \mapsto \tanh(v)$,
- Optimal discriminator: $T^*(\boldsymbol{x}) = \frac{1}{2}\mathrm{sign}\left(\frac{p(\boldsymbol{x})}{\widehat{p}(\boldsymbol{x})} - 1\right)$.

**PR-Divergence:**

- Notation: $\mathcal{D}_{\lambda\text{-PR}}$ for $\lambda > 0$,
- Generator Function: $f(u) = \max(\lambda u, 1) - \max(\lambda, 1)$,
- Convex Conjugate domain: $\mathrm{dom}(f^*) = [0, \lambda]$,
- Convex Conjugate: $f^*_\lambda(t) = t/\lambda$ for $\lambda \leq 1$ and $f^*_\lambda(t) = t/\lambda + \lambda - 1$ otherwise,
- Activation: $v \mapsto \lambda\sigma(v)$, where $\sigma$ is the sigmoid function,
- Optimal discriminator: $T^*(\boldsymbol{x}) = \lambda\mathrm{sign}\left(\frac{p(\boldsymbol{x})}{\widehat{p}(\boldsymbol{x})} - 1\right)$.

### A.2 Bregman Divernce

The Bregman Divergence under a strictly convex function $f : \mathbb{R}^n \to \mathbb{R}$ with a continuously differentiable interior, between two points $\boldsymbol{x}$ and $\boldsymbol{y}$ in the interior of the domain of $f$, is denoted by $\mathrm{Breg}_f$ and is defined as:

$$\mathrm{Breg}_f(\boldsymbol{x}, \boldsymbol{y}) = f(\boldsymbol{x}) - f(\boldsymbol{y}) - \langle \nabla f(\boldsymbol{y}), \boldsymbol{x} - \boldsymbol{y}\rangle$$

where $\nabla f(\boldsymbol{y})$ is the gradient of $f$ at $\boldsymbol{y}$, and $\langle ., . \rangle$ is the inner product in $\mathbb{R}^n$. It follows some properties as a distance metrics:

- **Non-Negativity:** For any $\boldsymbol{x}, \boldsymbol{y}$ in the interior of the domain of $f$, we have $\mathrm{Breg}_f(\boldsymbol{x}, \boldsymbol{y}) \geq 0$. The equality holds if and only if $\boldsymbol{x} = \boldsymbol{y}$.

- **Convexity:** The Bregman Divergence $\text{Breg}_f(\boldsymbol{x}, \boldsymbol{y})$ is convex in its first argument. That is, for any $\boldsymbol{x}_1, \boldsymbol{x}_2, \boldsymbol{y}$ in the interior of the domain of $f$ and any $t \in [0, 1]$, we have:

$$\text{Breg}_f(t\boldsymbol{x}_1 + (1-t)\boldsymbol{x}_2, \boldsymbol{y}) \le t\text{Breg}_f(\boldsymbol{x}_1, \boldsymbol{y}) + (1-t)\text{Breg}_f(\boldsymbol{x}_2, \boldsymbol{y})$$

- **Non-Symmetry:** Unlike some other distances or divergences, the Bregman Divergence is not symmetric, meaning that in general, $\text{Breg}_f(\boldsymbol{x}, \boldsymbol{y}) \neq \text{Breg}_f(\boldsymbol{y}, \boldsymbol{x})$.

- **Additivity:** If $f(\boldsymbol{x}) = f_1(x_1) + \cdots + f_d(x_d)$ where each $f_i$ is a convex function, then the Bregman Divergence decomposes into a sum of Bregman Divergences, i.e., $\text{Breg}_f(\boldsymbol{x}, \boldsymbol{y}) = \text{Breg}_{f_1}(x_1, y_1) + \cdots + \text{Breg}_{f_d}(x_d, y_d)$.

- **Taylor Approximation:** Bregman Divergence is essentially the error of the first-order Taylor approximation of $f$ around the point $\boldsymbol{y}$ at the point $\boldsymbol{x}$.

- **Connection with Dual Functions:** If $f$ is convex, then it has a convex conjugate (or dual function) $f^*$. The Bregman Divergence $\text{Breg}_f(\boldsymbol{x}, \boldsymbol{y})$ is related to the Bregman Divergence $\text{Breg}_{f^*}(\nabla f(\boldsymbol{y}), \nabla f(\boldsymbol{x}))$ between the gradients of $f$ at $\boldsymbol{x}$ and $\boldsymbol{y}$, where the gradient map $\nabla f$ serves as a Legendre transformation between the primal and dual spaces.

# B Proof and supplementary for Section 4

## B.1 Proof for Theorem 4.3

We have to prove that $\alpha(\lambda)$ can be written as a function of an $f$-divergence for any $\lambda \in \mathbb{R}^+$. First we can develop the expression of $\alpha(\lambda)$:

$$\alpha(\lambda) = \int_{\mathcal{X}} \min\left(\lambda p(\boldsymbol{x}), \widehat{p}(\boldsymbol{x})\right) \mathrm{d}\boldsymbol{x} \tag{16}$$

$$= \int_{\mathcal{X}} \widehat{p}(\boldsymbol{x}) \min\left(\lambda \frac{p(\boldsymbol{x})}{\widehat{p}(\boldsymbol{x})}, 1\right) \mathrm{d}\boldsymbol{x} \tag{17}$$

For this integral to be considered as an $f$-divergence, we need $f$ to be first convex lower semi-continuous and then to satisfy $f(1) = 0$. However, for every $a, b \in \mathbb{R}$, the min satisfies $\min(a, b) = a + b - \max(a, b)$. Therefore,

$$\alpha(\lambda) = \int_{\mathcal{X}} \widehat{p}(\boldsymbol{x}) \left[\lambda \frac{p(\boldsymbol{x})}{\widehat{p}(\boldsymbol{x})} + 1 - \max\left(\lambda \frac{p(\boldsymbol{x})}{\widehat{p}(\boldsymbol{x})}, 1\right)\right] \mathrm{d}\boldsymbol{x} \tag{18}$$

$$= \lambda \int_{\mathcal{X}} p(\boldsymbol{x}) \mathrm{d}\boldsymbol{x} + 1 - \int_{\mathcal{X}} \max\left(\lambda \frac{p(\boldsymbol{x})}{\widehat{p}(\boldsymbol{x})}, 1\right) \mathrm{d}\boldsymbol{x} \tag{19}$$

$$= \lambda + 1 - \int_{\mathcal{X}} \widehat{p}(\boldsymbol{x}) \max\left(\lambda \frac{p(\boldsymbol{x})}{\widehat{p}(\boldsymbol{x})}, 1\right) \mathrm{d}\boldsymbol{x} \tag{20}$$

Thus, we can take $f(u) = \max(\lambda u, 1) - \max(\lambda, 1)$ such that $f(1) = 0$. The precision becomes:

$$\alpha(\lambda) = \lambda + 1 - \int_{\mathcal{X}} \widehat{p}(\boldsymbol{x}) f\left(\frac{p(\boldsymbol{x})}{\widehat{p}(\boldsymbol{x})}\right) - \max(\lambda, 1) \int_{\mathcal{X}} \widehat{p}(\boldsymbol{x}) \mathrm{d}\boldsymbol{x} \tag{21}$$

$$= \min(\lambda, 1) - \int_{\mathcal{X}} \widehat{p}(\boldsymbol{x}) f\left(\frac{p(\boldsymbol{x})}{\widehat{p}(\boldsymbol{x})}\right) \mathrm{d}\boldsymbol{x} = \min(\lambda, 1) - \mathcal{D}_{\lambda\text{-PR}}(P, \widehat{P}). \tag{22}$$

Consequently, $\alpha(\lambda)$ can be written as a function of an $f$-divergence $\mathcal{D}_{\lambda\text{-PR}}$ with $f(u) = \max(\lambda u, 1) - \max(\lambda, 1)$.

Now we prove the converse. Suppose there exists a strictly decreasing **linear**[4] function $h : [0, 1] \to \mathbb{R}^+$ and an $f$-divergence $\mathcal{D}_f$ such that $h(\alpha_\lambda(P \| \widehat{P})) = \mathcal{D}_f(P \| \widehat{P})$ for all $P, \widehat{P} \in \mathcal{P}(\mathcal{X})$.

For $P = \widehat{P}$, we get from the definition of $\alpha_\lambda$ that $\alpha_\lambda(P \| P) = \min(\lambda, 1)$. Hence,

$$0 = \mathcal{D}_f(P \| P) = h(\alpha_\lambda(P \| P)) = h(\min(\lambda, 1)).$$

---

[4]We omitted the critical constraint of $h$ being linear in the original statement of the theorem in our submission by mistake. We apologize for this oversight. We stress again that the first part of the Theorem, which is the most important part, remains completely unaffected.

Combining the above with the fact that $h$ is a strictly decreasing linear function, we see that for any fixed $\lambda$, $h$ must be of the form, $h(u) = c_\lambda(\min(\lambda, 1) - u)$, where $c_\lambda > 0$ is a constant. Now,

$$\mathcal{D}_f(P\|\widehat{P}) = h(\alpha_\lambda(P\|\ \widehat{P})) = c_\lambda(\min(\lambda, 1) - \alpha_\lambda(P\|\ \widehat{P})) = c_\lambda \mathcal{D}_{\lambda\text{-PR}}(P\|\widehat{P}),$$

where the last equality follows from the first part of the theorem, which shows that $\alpha_\lambda(P\|\ \widehat{P}) = \min(\lambda, 1) - \mathcal{D}_{\lambda\text{-PR}}(P\|\widehat{P})$. Rewriting the above inequality, we get the following.

$$\mathcal{D}_{\lambda\text{-PR}}(P\|\widehat{P}) = \frac{1}{c_\lambda}\mathcal{D}_f(P\|\widehat{P}) = \mathcal{D}_{\frac{1}{c_\lambda}f}(P\|\widehat{P}).$$

By the uniqueness theorem of $f$-divergence $f(u) = \frac{c_1}{c_\lambda}f_\lambda(u) + c_2(u-1)$ for some constants $c_1, c_2 \in \mathbb{R}$.

## B.2   Proof of Proposition 4.2

If the generator function $f$ of the Precision-Recall Divergence is $f(u) = \max(\lambda u, 1) - \max(\lambda, 1)$ then its Fenchel conjugate function is:

$$f^*(t) = \sup_{u \in \text{dom}(f)} \{tu - f(u)\} = \max(\lambda, 1) + \sup_{u \in \mathbb{R}^+} \{tu - \max(\lambda u, 1)\} \tag{23}$$

If $t > \lambda$ or $\lambda < 0$, then the $\sup_{u \in \mathbb{R}^+}\{tu - \max(\lambda u, 1)\} = \infty$ for respectively $u \to \infty$ and $u \to -\infty$. The domain of $f^*$ is thus restricted to $[0, \lambda]$. Thus for $0 \leq t \leq \lambda$, the supremum is obtained for $u = 1/\lambda$ since $0$ is in the sub-differential of the function in $1/\lambda$ as Figure 6(b).

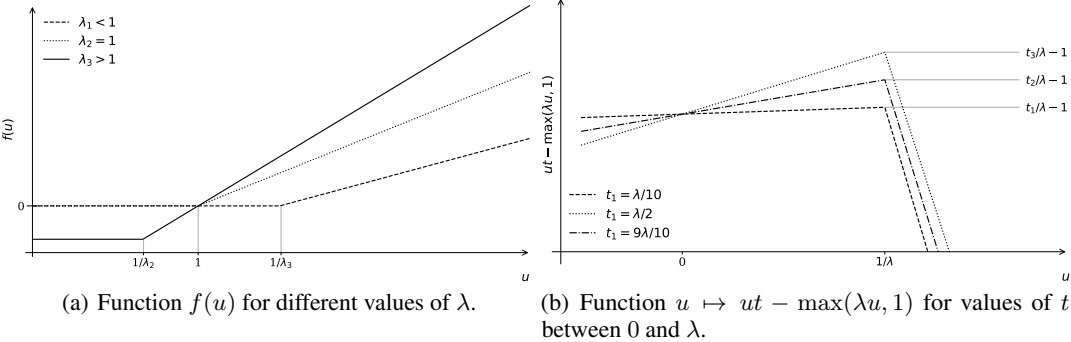

(a) Function $f(u)$ for different values of $\lambda$.

(b) Function $u \mapsto ut - \max(\lambda u, 1)$ for values of $t$ between $0$ and $\lambda$.

Figure B.6: Graphical illustration of $f_\lambda$ and $f_\lambda^*$. Both are piece-wise linear

Consequently the Fenchel conjugate of $f$ is:

$$\forall t \in [0, \lambda], \quad f^*(t) = \max(\lambda, 1) + t\lambda - 1 = \begin{cases} t/\lambda & \text{if } \lambda \leq 1, \\ t/\lambda - 1 + \lambda & \text{otherwise.} \end{cases} \tag{24}$$

Finally, the optimal discriminator $T^{\text{opt}}$ by taking the derivative of $f$ in $\frac{p(\boldsymbol{x})}{\widehat{p}(\boldsymbol{x})}$, we get:

$$T^{\text{opt}}(\boldsymbol{x}) = \nabla f\left(\frac{p(\boldsymbol{x})}{\widehat{p}(\boldsymbol{x})}\right) = \begin{cases} \lambda & \text{if } \frac{p(\boldsymbol{x})}{\widehat{p}(\boldsymbol{x})} \leq 1/\lambda, \\ 0 & \text{otherwise.} \end{cases} \tag{25}$$

Then we can compute the compute the reverse $\mathcal{D}_{\lambda\text{-PR}}$:

$$\mathcal{D}_{\lambda\text{-PR}}(\widehat{P}\|P) = \int_{\mathcal{X}} p(\boldsymbol{x}) f_\lambda \left( \frac{\widehat{p}(\boldsymbol{x})}{p(\boldsymbol{x})} \right) \mathrm{d}\boldsymbol{x} \tag{26}$$

$$= \int_{\mathcal{X}} \max(\lambda \widehat{p}(\boldsymbol{x}), p(\boldsymbol{x})) - p(\boldsymbol{x}) \max(\lambda, 1) \, \mathrm{d}\boldsymbol{x} \tag{27}$$

$$= \lambda \left( \int_{\mathcal{X}} \max(\widehat{p}(\boldsymbol{x}), p(\boldsymbol{x})/\lambda) \mathrm{d}\boldsymbol{x} - \max(1, 1/\lambda) \right) \tag{28}$$

$$= \lambda \int_{\mathcal{X}} \widehat{p}(\boldsymbol{x}) \max(1, \frac{p(\boldsymbol{x})}{\widehat{p}(\boldsymbol{x})}/\lambda) - \widehat{p}(\boldsymbol{x}) \max(1, 1/\lambda) \, \mathrm{d}\boldsymbol{x} \tag{29}$$

$$= \lambda \int_{\mathcal{X}} \widehat{p}(\boldsymbol{x}) f_{1/\lambda} \left( \frac{p(\boldsymbol{x})}{\widehat{p}(\boldsymbol{x})} \right) \mathrm{d}\boldsymbol{x} \tag{30}$$

$$= \lambda \mathcal{D}_{\frac{1}{\lambda}\text{-PR}}(P\|\widehat{P}). \tag{31}$$

With this results, we can show that :

$$\mathcal{D}_{\text{TV}}(P\|\widehat{P}) = \int_{\mathcal{X}} |p(\boldsymbol{x}) - \widehat{p}(\boldsymbol{x})| \, \mathrm{d}\boldsymbol{x} \tag{32}$$

$$= \int_{\mathcal{X}} \max(p(\boldsymbol{x}) - \widehat{p}(\boldsymbol{x}), 0) + \max(\widehat{p}(\boldsymbol{x}) - p(\boldsymbol{x}), 0) \mathrm{d}\boldsymbol{x} \tag{33}$$

Then since $\mathcal{D}_{1\text{-PR}}(P\|\widehat{P}) = \int_{\mathcal{X}} \max(\widehat{p}(\boldsymbol{x}), p(\boldsymbol{x})) - p(\boldsymbol{x}) \mathrm{d}\boldsymbol{x} = \int_{\mathcal{X}} \max(\widehat{p}(\boldsymbol{x}) - p(\boldsymbol{x}), 0) \mathrm{d}\boldsymbol{x}$ and $\mathcal{D}_{1\text{-PR}}(P\|\widehat{P}) = \mathcal{D}_{1\text{-PR}}(\widehat{P}\|P)$, we have:

$$\mathcal{D}_{\text{TV}}(P\|\widehat{P}) = \mathcal{D}_{1\text{-PR}}(P\|\widehat{P}) + \mathcal{D}_{1\text{-PR}}(\widehat{P}\|P) \tag{34}$$

$$= 2\mathcal{D}_{1\text{-PR}}(P\|\widehat{P}). \tag{35}$$

## B.3    Proof of Theorem 5.2

From now on, assume the support of $P$ and $\widehat{P}$ coincide. For any $T : \mathcal{X} \to \mathbb{R}$,

$$\mathcal{D}_{f,T}^{\text{dual}}(P\|\widehat{P}) = \mathbb{E}_{x\sim P}[d(x)] - \mathbb{E}_{x\sim Q}[f^*(d(x))]$$

$$= \mathbb{E}_{x\sim Q}\left[ \frac{p(\boldsymbol{x})}{\widehat{p}(\boldsymbol{x})} d(x) - f^*(d(x)) \right]$$

Let $T^{\text{opt}} \in \arg\sup \mathcal{D}_{f,T}^{\text{dual}}(P\|\widehat{P})$. For any $T : \mathcal{X} \to \mathbb{R}$

$$\mathcal{D}_f(P\|\widehat{P}) - D_{f,T}^{\text{dual}}(P\|\widehat{P}) = \mathcal{D}_{f,T^{\text{opt}}}^{\text{dual}}(P\|\widehat{P}) - \mathcal{D}_{f,T}^{\text{dual}}(P\|\widehat{P})$$

$$= \mathbb{E}_{\widehat{P}}\left[ \frac{p(\boldsymbol{x})}{\widehat{p}(\boldsymbol{x})} \left( T^{\text{opt}}(\boldsymbol{x}) - T(\boldsymbol{x}) \right) - f^*\left( T^{\text{opt}}(\boldsymbol{x}) \right) + f^*\left( T(\boldsymbol{x}) \right) \right]$$

It is known that for all $x \in \mathcal{X}$ we have $\nabla f^*(T^{\text{opt}}(\boldsymbol{x})) = \frac{p(\boldsymbol{x})}{\widehat{p}(\boldsymbol{x})}$:

$$\mathcal{D}_f(P\|\widehat{P}) - \mathcal{D}_{f,T}^{\text{dual}}(P\|\widehat{P}) = \mathbb{E}_{\widehat{P}}\left[ \nabla f^*(T^{\text{opt}}(\boldsymbol{x})) \left( T^{\text{opt}}(\boldsymbol{x}) - T(\boldsymbol{x}) \right) - f^*\left( T^{\text{opt}}(\boldsymbol{x}) \right) + f^*\left( T(\boldsymbol{x}) \right) \right]$$

Recall that for any continuously differentiable strictly convex function $f$, the Bregman divergence of $f$ is $\text{Breg}_f(a,b) = f(a) - f(b) - \langle \nabla f(b), a - b \rangle$. So we have

$$\mathcal{D}_f(P\|\widehat{P}) - \mathcal{D}_{f,T}^{\text{dual}}(P\|\widehat{P}) = \mathbb{E}_{\widehat{P}}\left[ \text{Breg}_{f^*}\left( T(\boldsymbol{x}), T^{\text{opt}}(\boldsymbol{x}) \right) \right]$$

Let us now use the following property: $\text{Breg}_f(a,b) = \text{Breg}_{f^*}(a^*, b^*)$ where $a^* = \nabla f(a)$ and $b^* = \nabla f(b)$.

$$\mathcal{D}_f(P\|\widehat{P}) - \mathcal{D}_{f,T}^{\text{dual}}(P\|\widehat{P}) = \mathbb{E}_{\widehat{P}}\left[\text{Breg}_f\left(\nabla f^*(T(\boldsymbol{x})), \nabla f^*(T^{\text{opt}}(\boldsymbol{x})))\right)\right]$$

$$= \mathbb{E}_{\widehat{P}}\left[\text{Breg}_f\left(\nabla f^*(T(\boldsymbol{x})), \frac{p(\boldsymbol{x})}{\widehat{p}(\boldsymbol{x})}\right)\right]$$

Let us define $r(\boldsymbol{x}) = \nabla f^* T(\boldsymbol{x})$ as our estimator of $p(\boldsymbol{x})/\widehat{p}(\boldsymbol{x})$. So finally, we have

$$\mathcal{D}_f(P\|\widehat{P}) - \mathcal{D}_{f,T}^{\text{dual}}(P\|\widehat{P}) = \mathbb{E}_{\widehat{P}}\left[\text{Breg}_f\left(r(\boldsymbol{x}), \frac{p(\boldsymbol{x})}{\widehat{p}(\boldsymbol{x})}\right)\right]$$

### B.4 Proof of Theorem 5.3

Now assume that $f$ is $\mu$-strongly convex, then $\text{Breg}_f(a,b) \geq \frac{\mu}{2}\|a-b\|^2$ If $\mathbb{E}_{\widehat{P}}\left[\text{Breg}_f\left(r(\boldsymbol{x}), \frac{p(\boldsymbol{x})}{\widehat{p}(\boldsymbol{x})}\right)\right] \leq \epsilon$ and if $f$ is $\mu$-strongly convex, then

$$\mathbb{E}_{\widehat{P}}\left[\left(r(\boldsymbol{x}) - \frac{p(\boldsymbol{x})}{\widehat{p}(\boldsymbol{x})}\right)^2\right] \leq \frac{2\epsilon}{\mu}. \tag{36}$$

Consider an arbitrary f-divergence $\mathcal{D}_g(P\|\widehat{P}) = \int g\left(\frac{dP}{d\widehat{P}}\right) d\widehat{P}$. Define $\mathcal{D}_{g,T}^{\text{primal}}(P\|\widehat{P}) = \int g(r(\boldsymbol{x})) d\widehat{P}$. Then,

$$\left|\mathcal{D}_g(P\|\widehat{P}) - \mathcal{D}_{g,T}^{\text{primal}}(P\|\widehat{P})\right| = \left|\mathbb{E}_{\widehat{P}}\left[g\left(\frac{p(\boldsymbol{x})}{\widehat{p}(\boldsymbol{x})}\right) - g(r(\boldsymbol{x}))\right]\right|$$

$$\leq \mathbb{E}_{\widehat{P}}\left[\left|g\left(\frac{p(\boldsymbol{x})}{\widehat{p}(\boldsymbol{x})}\right) - g(r(\boldsymbol{x}))\right|\right]$$

$$\overset{(a)}{\leq} \mathbb{E}_{\widehat{P}}\left[\sigma |e(x)|\right]$$

$$= \sigma \mathbb{E}_{\widehat{P}}\left[|e(x)|\right]$$

$$\overset{(b)}{\leq} \sigma\sqrt{\mathbb{E}_{\widehat{P}}\left[e(x)^2\right]}$$

$$\overset{(c)}{\leq} \sigma\sqrt{\frac{2\epsilon}{\mu}},$$

where $(a)$ follows from the $\sigma$-Lipschitz assumption on $g$, $(b)$ follows from Jensen's inequality and finally, $(c)$ follows from equation (36). s

### B.5 Proof of Theorem 8

Let $c : \mathbb{R}^+ \mapsto \mathbb{R}$ be a $\mathcal{C}^2$ function and take $u_{\min}$ and $u_{\max}$. The goal is to express $f(u)$ for all $u \in [u_{\min}, u_{\max}]$ as a weighted average of $f_\lambda^{\text{PR}}$:

$$\forall u \in \mathbb{R}_*^+, \int_{1/u_{\max}}^{1/u_{\min}} c''(\lambda) f_\lambda^{\text{PR}}(u) d\lambda = \int_{1/u_{\max}}^{1/u_{\min}} c''(\lambda)\left[\max(\lambda u, 1) - \max(\lambda, 1)\right] d\lambda \tag{37}$$

First, let us assume that $u_{\min} \leq 1$ and $u_{\max} \geq 1$, then the terms can be decomposed and the integral split to evaluate the max:

$$\int_{1/u_{\max}}^{1/u_{\min}} c''(\lambda) f_\lambda^{\mathrm{PR}}(u)\mathrm{d}\lambda = \int_{1/u_{\max}}^{1/u_{\min}} c''(\lambda) \max(\lambda u, 1)\mathrm{d}\lambda - \int_{1/u_{\max}}^{1/u_{\min}} c''(\lambda) \max(\lambda, 1)\,\mathrm{d}\lambda \quad (38)$$

$$= \int_{1/u_{\max}}^{1/u} c''(\lambda) \max(\lambda u, 1)\mathrm{d}\lambda + \int_{1/u}^{1/u_{\min}} c''(\lambda) \max(\lambda u, 1)\mathrm{d}\lambda$$

$$- \int_{1/u_{\max}}^{1} c''(\lambda) \max(\lambda, 1)\,\mathrm{d}\lambda - \int_{1}^{1/u_{\min}} c''(\lambda) \max(\lambda, 1)\,\mathrm{d}\lambda$$

$$(39)$$

$$= \int_{1/u_{\max}}^{1/u} c''(\lambda)\mathrm{d}\lambda + \int_{1/u}^{1/u_{\min}} c''(\lambda)\lambda u\,\mathrm{d}\lambda - \int_{1/u_{\max}}^{1} c''(\lambda)\mathrm{d}\lambda - \int_{1}^{1/u_{\min}} c''(\lambda)\lambda\,\mathrm{d}\lambda.$$

$$(40)$$

By integrating by parts, we have: $\int_{1/u_{\max}}^{1/u_{\min}} c''(\lambda)\lambda\,\mathrm{d}\lambda = [c'(\lambda)\lambda]_{1/u_{\max}}^{1/u_{\min}} - \int_{1/u_{\max}}^{1/u_{\min}} c'(\lambda)\mathrm{d}\lambda$ so it satisfies:

$$\int_{1/u_{\max}}^{1/u_{\min}} c''(\lambda) f_\lambda^{\mathrm{PR}}(u)\mathrm{d}\lambda = \int_{1/u_{\max}}^{1/u} c''(\lambda)\mathrm{d}\lambda + u\,[c'(\lambda)\lambda]_{1/u}^{1/u_{\min}} - u\int_{1/u}^{1/u_{\min}} c'(\lambda)\mathrm{d}\lambda \quad (41)$$

$$- \int_{1/u_{\max}}^{1} c''(\lambda)\mathrm{d}\lambda - [c'(\lambda)\lambda]_1^{1/u_{\min}} + \int_{1}^{1/u_{\min}} c'(\lambda)\mathrm{d}\lambda$$

$$= [c'(\lambda)]_{1/u_{\max}}^{1/u} + u\,[c'(\lambda)\lambda]_{1/u}^{1/u_{\min}} - u\,[c(\lambda)]_{1/u}^{1/u_{\min}} \quad (42)$$

$$- [c'(\lambda)]_{1/u_{\max}}^{1} - [c'(\lambda)\lambda]_1^{1/u_{\min}} + [c(\lambda)]_1^{1/u_{\min}}$$

$$= c'\left(\frac{1}{u}\right) - c'(0) + uc'\left(\frac{1}{u_{\min}}\right) - uc'\left(\frac{1}{u}\right)\frac{1}{u} - uc\left(\frac{1}{u_{\min}}\right) + uc\left(\frac{1}{u}\right) - c'(1)$$

$$+ c'(0) - c'\left(\frac{1}{u_{\min}}\right)\frac{1}{u_{\min}} + c'(1) \times 1 + c\left(\frac{1}{u_{\min}}\right) - c(1)$$

$$(43)$$

$$= \left[c'\left(\frac{1}{u_{\min}}\right)\frac{1}{u_{\min}} - c\left(\frac{1}{u_{\min}}\right)\right](u-1) + uc\left(\frac{1}{u}\right) - c(1). \quad (44)$$

We would like $\int_{1/u_{\max}}^{1/u_{\min}} c''(\lambda) f_\lambda^{\mathrm{PR}}(u)\mathrm{d}\lambda$ to be equal to $f$ between $u_{\min}$ and $u_{\max}$. But since two $f$-divergences generated by $f$ and $g$ are equals if there is a $c \in \mathbb{R}$ such that $f(u) = g(u) + c(u - 1)$, the Divergence generated by $\int_{1/u_{\max}}^{1/u_{\min}} c''(\lambda) f_\lambda^{\mathrm{PR}}(u)\mathrm{d}\lambda$ is equal to the divergence generated by $uc\left(\frac{1}{u}\right) - c(1)$. Therefore, we require $c$ to satisfy:

$$\forall u \in [u_{\min}, u_{\max}], \quad f(u) = uc\left(\frac{1}{u}\right) - c(1).$$

By differentiating with respect to $u$, we have:

$$f'(u) = \lim_{\lambda \to \infty} [c'(\lambda)\lambda - c(\lambda)] + c\left(\frac{1}{u}\right) - \frac{1}{u}c'\left(\frac{1}{u}\right). \quad (45)$$

And finally:

$$f''(u) = -\frac{1}{u^2}c\left(\frac{1}{u}\right) + \frac{1}{u^2}c'\left(\frac{1}{u}\right) + \frac{1}{u^3}c''\left(\frac{1}{u}\right) \quad (46)$$

$$= \frac{1}{u^3}c''\left(\frac{1}{u}\right). \quad (47)$$

Consequently, with $\lambda = 1/u$, we have that:

$$c''(\lambda) = \frac{1}{\lambda^3}f''\left(\frac{1}{\lambda}\right). \quad (48)$$

With such a results, with $m = \min_{\mathcal{X}}(\frac{\widehat{p}(\boldsymbol{x})}{p(\boldsymbol{x})})$ and $M = \max_{\mathcal{X}}(\frac{\widehat{p}(\boldsymbol{x})}{p(\boldsymbol{x})})$, we can write any $f$-divergence as:

$$
\begin{aligned}
\mathcal{D}_f(P\|\widehat{P}) &= \int_{\mathcal{X}} \widehat{p}(\boldsymbol{x}) f\left(\frac{p(\boldsymbol{x})}{\widehat{p}(\boldsymbol{x})}\right) \mathrm{d}\boldsymbol{x} \\
&= \int_{\mathcal{X}} \widehat{p}(\boldsymbol{x}) \int_m^M \frac{1}{\lambda^3} f''\left(\frac{1}{\lambda}\right) f_\lambda^{\mathrm{PR}}\left(\frac{p(\boldsymbol{x})}{\widehat{p}(\boldsymbol{x})}\right) \mathrm{d}\lambda \mathrm{d}\boldsymbol{x} \\
&= \int_m^M \int_{\mathcal{X}} \frac{1}{\lambda^3} f''\left(\frac{1}{\lambda}\right) \widehat{p}(\boldsymbol{x}) f_\lambda^{\mathrm{PR}}\left(\frac{p(\boldsymbol{x})}{\widehat{p}(\boldsymbol{x})}\right) \mathrm{d}\lambda \mathrm{d}\boldsymbol{x} \\
&= \int_m^M \frac{1}{\lambda^3} f''\left(\frac{1}{\lambda}\right) \left[\int_{\mathcal{X}} \widehat{p}(\boldsymbol{x}) f_\lambda^{\mathrm{PR}}\left(\frac{p(\boldsymbol{x})}{\widehat{p}(\boldsymbol{x})}\right) \mathrm{d}\boldsymbol{x}\right] \mathrm{d}\lambda \\
&= \int_m^M \frac{1}{\lambda^3} f''\left(\frac{1}{\lambda}\right) \mathcal{D}_{\lambda\text{-PR}}(P\|\widehat{P}) \mathrm{d}\lambda
\end{aligned}
$$

### B.6 Proof of Corollary 4.5

In particular for the $\mathcal{D}_{\mathrm{KL}}$, $f(u) = u \log u$, therefore $f''(u) = 1/u$ which gives:

$$
\mathcal{D}_{\mathrm{KL}}(P\|\widehat{P}) = \int_m^M \frac{1}{\lambda^2} \mathcal{D}_{\lambda\text{-PR}}(P\|\widehat{P}) \mathrm{d}\lambda. \tag{49}
$$

And for the $\mathcal{D}_{\mathrm{rKL}}$ we can either use Equation 48 with $f(u) = -\log u$ or use the fact that $\mathcal{D}_{\lambda\text{-PR}}(P\|\widehat{P}) = \lambda \mathcal{D}_{1/\lambda\text{-PR}}(\widehat{P}\|P)$:

$$
\mathcal{D}_{\mathrm{rKL}}(P\|\widehat{P}) = \int_m^M \frac{1}{\lambda} \mathcal{D}_{\lambda\text{-PR}}(P\|\widehat{P}) \mathrm{d}\lambda. \tag{50}
$$

## C  Minimizing the Precision-Recall divergence

In Section 5, we outline a novel strategy to train models for challenging $f$-divergences. Rather than directly employing the $f$-GAN framework and minimizing the dual variational form of $\mathcal{D}_{\lambda\text{-PR}}$, we introduce an auxiliary function, denoted by $g$, which is easier to train. Practical choices for $g$ could include functions such as $f_{\mathrm{KL}}$ or $f_{\chi^2}$, which have been shown to be easily trainable.

Using this approach, a discriminator $T$ is trained with the objective of maximizing $\mathcal{D}_{g,T}^{\mathrm{dual}}$. This is achieved by drawing samples from both $P$ and $\widehat{P}$ and computing the estimate in the dual form. Under optimal conditions, we find $\nabla g^*(T^{\mathrm{opt}}(\boldsymbol{x})) = p(\boldsymbol{x})/\widehat{p}(\boldsymbol{x})$, enabling us to estimate the $f$-divergence based on $T$ using what we term the primal estimate.

The primal estimate is defined as $\mathcal{D}_{f,T}^{\mathrm{primal}}(P\|\widehat{P}) = \int_{\mathcal{X}} \widehat{p}(\boldsymbol{x}) f\left(r(\boldsymbol{x})\right) \mathrm{d}\boldsymbol{x}$, where $r(\boldsymbol{x}) = \nabla g^*(T(\boldsymbol{x}))$. This strategy and its corresponding training procedure are visually represented in Figure C.7.

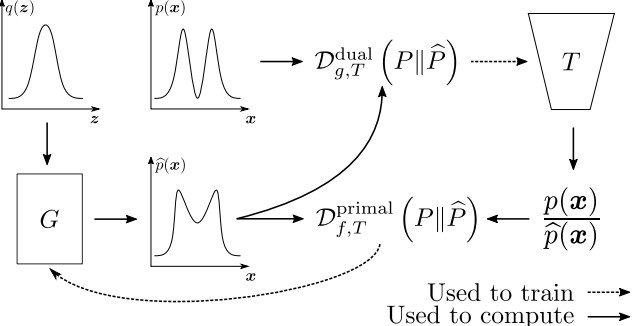

Figure C.7: Training procedure: the discriminator $T$ is trained based on $\mathcal{D}_{g,T}^{\mathrm{dual}}$ and $G$ is trained on $\mathcal{D}^{\mathrm{primal}}f, T$.

We have shown that the choice of $g$ affects the quality of the estimation of $\mathcal{D}_f$. We have shown that to have guarantees on the quality of the estimation, $g$ must be strictly convex. We show empirically that $g = f_{\chi^2}$ leads to a better approximation than $g = f_{\mathrm{KL}}$. We train 200 discriminators on 20 Gaussian random two-dimensional mixtures to maximize $\mathcal{D}_{g,T}^{\mathrm{dual}}$ with $g = f_{\mathrm{KL}}$ and $g = f_{\chi^2}$. In Figure C.8, we report the results. On the $x$-axis, we report the *training* loss quality $\mathcal{D}_g(P\|\widehat{P}) - \mathcal{D}_{g,T}^{\mathrm{dual}}(P\|\widehat{P})$ and on the $y$-axis we report the *estimation quality* : $\mathcal{D}_f(P\|\widehat{P}) - \mathcal{D}_{f,T}^{\mathrm{primal}}(P\|\widehat{P})|$. The dotted lines correspond to the Mahalanobis distance of each set of experiments. In this experiment, we use a 4 linear layers 2-1024-512-256-1 neural network with LeakyRelu activation between layers. The last activation is set to match $\mathrm{dom}(f^*)$ (see Section A.1). We use a learning rate of $2.10^{-5}$ with Adam optimizer.

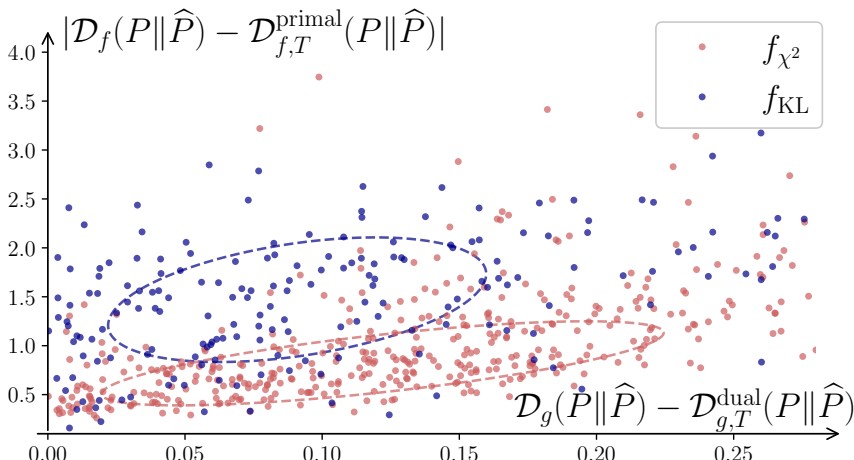

Figure C.8: We compare the quality of the primal estimation using two discriminators, one trained using the $\mathcal{D}_{\mathrm{KL}}$ and $\mathcal{D}_{\chi^2}$ over 200 experiments. The lower the point is on the $y$-axis, the better the estimate. As we can see, even distant dual estimates of the $f_{\chi^2}$ (points on the right side of the chart) tend to provide better estimates of the primal.

We observe in this experiment that 1) the difference between $\mathcal{D}_{\mathrm{KL}}$ and its dual is usually lower (the blue circle is lower on $x$-axis) and yet 2) the estimation of $\mathcal{D}_f$ is usually better with $g = f_{\chi^2}$ (the red circle is lower on $y$-axis). These observations corroborate our claim that employing $f_{\chi^2}$ as the auxiliary divergence is a more advantageous choice compared to $f_{\mathrm{KL}}$. Although both divergence measures exhibit satisfactory performance in the context of the MNIST and FashionMNIST experiments, we find $f_{\chi^2}$ to demonstrate superior stability, reinforcing its suitability in this training framework.

**Complete version of the algorithm**  In Section 5, we introduce a condensed version of our training algorithm. For practical purposes, however, we implement a more refined variant of this algorithm using stochastic gradient descent, detailed in Algorithm 1. This approach involves segmenting the dataset into batches of size $N$ and computing the distinct losses for each individual batch. Our algorithm is closely related to the GAN training procedure, operating on an iterative principle of alternating updates to the discriminator $T$ and the generator $G$ until a convergence state is reached. In particular, our method differs from the standard GAN training protocol in two key aspects. First, the generator $G$ in our case is trained based on an estimation of a distinct $f$-divergence. Second, this specific $f$-divergence is calculated in terms of its primal form, rather than its dual form, a marked departure from the established GAN training practice.

**Algorithm 1** Stochastic Gradient Descent for the two-step approach for minimizing $\mathcal{D}_{\lambda\text{-}\mathrm{PR}}$. For each batch $B$ composing the dataset $D$, the discriminator $T$ is trained on $\mathcal{L}_T$, the empirical estimation of $\mathcal{D}_{f,T}^{\mathrm{dual}}(P\|\widehat{P})$ and the generator $G$ is trained on $\mathcal{L}_G$, the empirical estimation of $\mathcal{D}_{\lambda\text{-}\mathrm{PR}}^{\mathrm{primal}}(P\|\widehat{P})$.

---

**Input:** Generator $G$, Discriminator $T$, Dataset $D$
$g^* \leftarrow f_{\chi^2}^*$ or $f_{\mathrm{KL}}^*$
$f \leftarrow f_\lambda$
**for** epoch $e = 1, \ldots, E$ **do**
   **for** $B \in D$ **do**
      $\mathcal{L}_T, \mathcal{L}_G \leftarrow 0, 0$
      **for** $\boldsymbol{x}_1^{\mathrm{real}}, \ldots, \boldsymbol{x}_N^{\mathrm{real}} \in B$ **do**
         **for** $i = 1$ to $N$ **do**
            Generate $\boldsymbol{x}_i^{\mathrm{fake}} = G(\boldsymbol{z})$ with $\boldsymbol{z} \sim \mathcal{N}(\boldsymbol{0}_d, \boldsymbol{I}_d)$
            $\mathcal{L}_T \leftarrow \mathcal{L}_T + T(\boldsymbol{x}_i^{\mathrm{real}}) - f^*(T(\boldsymbol{x}_i^{\mathrm{fake}}))$
         **end for**
         Update parameters of $T$ by ascending the gradient $\nabla \mathcal{L}_T$.
         **for** $i = 1$ to $N$ **do**
            $\mathcal{L}_G \leftarrow \mathcal{L}_G + g\left(\nabla f^*(T(\boldsymbol{x}_i^{\mathrm{fake}}))\right)$
         **end for**
         Update parameters of $G$ by descending the gradient $\nabla \mathcal{L}_G$.
      **end for**
   **end for**
**end for**

---

# D    Experiments

## D.1    Naive Approach

In this section, we juxtapose the straightforward method with our technique delineated in Section 5. We achieve this by evaluating the optimization process of two identical $G$ models: one is trained using the conventional approach and the other using our methodology. In reality, there exist two significant variations between these two training procedures: training losses and the final layer of the discriminator $T$. In both procedures, we use $f^*(T(\boldsymbol{x}))$, implying that the co-domain must be encapsulated within $\mathrm{dom}(f^*)$. More information on activation can be found in the Appendix A.1. Aside from these disparities, we utilize the identical training process as outlined in the previous section to train the models on CIFAR-10. The training procedures are compared in Figure D.9. To evaluate training procedures, we plot the loss $\mathcal{L}_G$, used to train $G$, either the dual estimation of $\mathcal{D}_{\lambda\text{-}\mathrm{PR}}$ in the naive approach or the primal estimation of $\mathcal{D}_{\lambda\text{-}\mathrm{PR}}$ in our method. To track the evolution of the discriminator, we plot discriminating predictions on the input conditioned on the class (real or fake). Precisely, we compute the accuracy on a batch of real or fake images, how many the discriminator achieves to identify as such. Finally, we plot the Precision and Recall computed at each epoch.

We observe that for varying values of $\lambda$, the discriminator $T$ undergoes the training process using the naive approach, leading to an unsatisfactory estimation of $\mathcal{D}_{f,T}^{\mathrm{dual}}$.

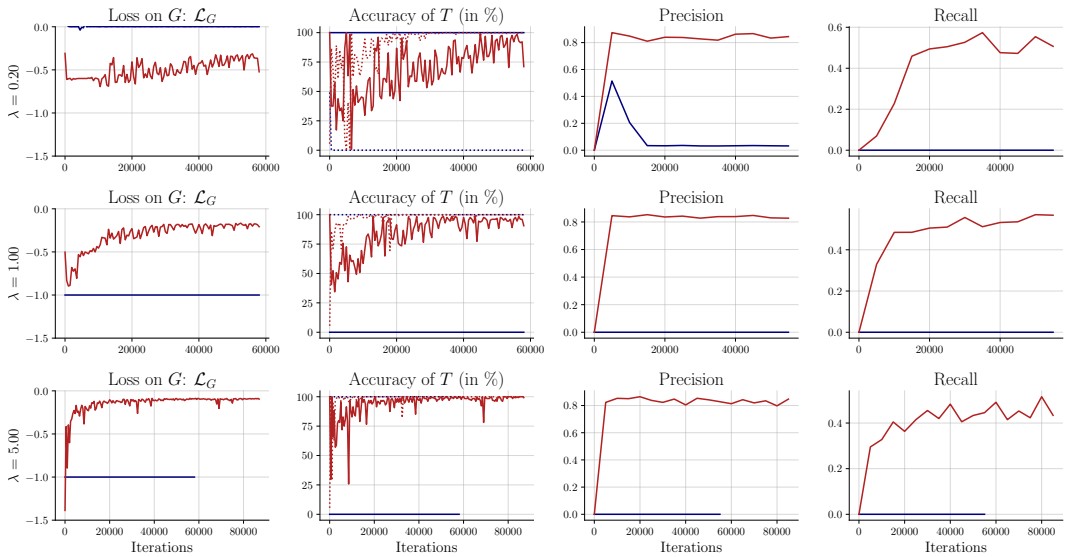

Figure D.9: Results for training BigGAN models on CIFAR-10 using the naive approach (in blue) and our method (in red). For $\lambda = 0.2$, $\lambda = 1$ and $\lambda = 5$, the loss represented by $G$ in learned. We obverse that for the naive approach, this loss is constant. Then, we plot the accuracy of the discriminator. We plot the accuracy conditioned on the class: in the dotted line the accuracy on the images of the dataset $\boldsymbol{x}^{\text{real}}$ and in the solid line the accuracy on the generated images $\boldsymbol{x}^{\text{fake}}$. Then we plot the Precision and Recall during training.

## D.2 Synthetic data: 8 Gaussians

For the synthetic data, we use a RealNVP [14] for the generator $G$. We use an 8-coupling step composed of each of 2 linear layers 2-256-2 with LeakyRelu activation in between. For the discriminator, we used a 4 linear layers 2-1024-512-256-1 neural network with LeakyRelu activation between layers. For both, we use Adam optimizer with a learning rate of $2.10^{-5}$ for $G$ and $1.10^{-4}$ for $T$. $G$ has 540k parameters and 660k for $T$.

Then, to estimate the PR curves, both methods from Sajjadi et al. [44] and Simon et al. [47] are developed for image dataset. To estimate these curves, we use our own estimation of $\mathcal{D}_{\lambda\text{-PR}}$. We computed $\mathcal{D}_{\lambda\text{-PR}}$ for $\lambda = \tan(\theta)$ with $\theta$ between 0 and $\pi/2$. The observations from the generated curves underscore a compelling finding: Each model has been effectively optimized for a unique, specific $\lambda$, resulting in its superior performance on the corresponding $\mathcal{D}_{\lambda\text{-PR}}$. This indicates a clear correspondence between the intended optimization and the achieved performance, suggesting the efficacy of our training algorithm in optimizing models for specific performance metrics. Such a successful matching of model optimization to $\lambda$ underscores the feasibility of tailoring model training according to a specific precision-recall trade-off objective.

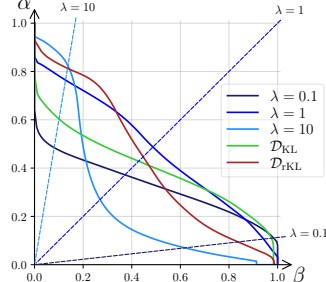

Figure D.10: PR Curves of the models in Figure 1. In shades of blue, the curves correspond to the model trained to minimize $\mathcal{D}_{\lambda\text{-PR}}$, with dark blue for $\lambda = 0.1$, medium blue for $\lambda = 1$, and light blue for $\lambda = 1$. The green curve corresponds to $\mathcal{D}_{\text{KL}}$ and the red curve to $\mathcal{D}_{\text{rKL}}$.

## D.3 MNIST and FashionMNIST

The training procedure for MNIST and FashionMNIST is strictly the same. For both we use a multiscale glow [27]. The model has three levels processing images of size $4 \times 16 \times 16$, $16 \times 8 \times 8$, $64 \times 8 \times 8$. Each level has 16 blocks of affine coupling with 3 layers of 512 channels of convolutional

operations, leading to a total of $85.2$M parameters. For the discriminator, we use a 4 linear layers 1024-1024-512-256-1 neural network with LeakyRelu activation between layers, with $1.7$M parameters. Both are trained with Adam using a learning rate of $1.10^{-5}$ for $T$ and $1.10^{-6}$ for $G$. For both dataset, we train a model for 250 epochs using maximum likelihood estimation (MLE) in 4 GPUs V100 ($\sim$ 200 hours). The models are then fine-tuned with their different losses on 12 V100 GPUs for 30 epochs ($\sim$ 2 hours). For two epochs we train the discriminator only and then both. For MNIST, we report results as graphs in Figure D.11, the quantitative results in Table 4 and samples in Figure D.3. For FashionMNIST, we report the results as graph in Figure D.13, the quantitative results in Table 4 and samples in Figure D.3.

Table 4: Quantitative evaluation of various Glow [27] models on the MNIST and FashionMNIST datasets. Models differ by their choice of $\mathcal{D}_f$, $\mathcal{D}_g$, and $\lambda$. The performance metrics include the FID ($\downarrow$), P ($\downarrow$) and R ($\uparrow$). The best model is highlighted in **bold**. FID is calculated for 50k samples. P and R are culculated for 10k samples and $k = 3$.

| | Model | | | MNIST | | | FashionMNIST | |
|---|---|---|---|---|---|---|---|---|
| $\mathcal{D}_g$ | $\mathcal{D}_f$ | $\lambda$ | FID | P | R | FID | P | R |
| | MLE | | 7.68 | 70.03 | 76.52 | 66.64 | 58.19 | 47.35 |
| $\mathcal{D}_{\mathrm{KL}}$ | $\mathcal{D}_{\mathrm{KL}}$ | - | 7.76 | 70.68 | **77.19** | 71.72 | 51.44 | 45.61 |
| $\mathcal{D}_{\mathrm{rKL}}$ | $\mathcal{D}_{\mathrm{rKL}}$ | - | 10.50 | 74.64 | 67.21 | 51.65 | 67.68 | 40.76 |
| $\mathcal{D}_{\chi^2}$ | $\mathcal{D}_{\lambda\text{-PR}}$ | 0.10 | 6.84 | 78.24 | 72.11 | 48.11 | 66.57 | 45.80 |
| $\mathcal{D}_{\chi^2}$ | $\mathcal{D}_{\lambda\text{-PR}}$ | 0.20 | 6.48 | 78.84 | 71.02 | **41.97** | 66.41 | 46.62 |
| $\mathcal{D}_{\chi^2}$ | $\mathcal{D}_{\lambda\text{-PR}}$ | 0.50 | 6.57 | 79.09 | 70.26 | 46.04 | 71.20 | 46.37 |
| $\mathcal{D}_{\chi^2}$ | $\mathcal{D}_{\lambda\text{-PR}}$ | 1.00 | 7.95 | 79.55 | 69.08 | 52.88 | 73.32 | 36.44 |
| $\mathcal{D}_{\chi^2}$ | $\mathcal{D}_{\lambda\text{-PR}}$ | 2.00 | 10.01 | 80.37 | 67.84 | 101.33 | **76.47** | 39.40 |
| $\mathcal{D}_{\chi^2}$ | $\mathcal{D}_{\lambda\text{-PR}}$ | 5.00 | 15.31 | 81.34 | 67.00 | 118.54 | 73.44 | 35.14 |
| $\mathcal{D}_{\chi^2}$ | $\mathcal{D}_{\lambda\text{-PR}}$ | 10.00 | 20.88 | 81.71 | 66.62 | 91.20 | 72.14 | 19.48 |
| $\mathcal{D}_{\mathrm{KL}}$ | $\mathcal{D}_{\lambda\text{-PR}}$ | 0.10 | 4.66 | 76.06 | 73.88 | 42.85 | 66.48 | **49.73** |
| $\mathcal{D}_{\mathrm{KL}}$ | $\mathcal{D}_{\lambda\text{-PR}}$ | 0.20 | **4.45** | 76.92 | 71.92 | 48.25 | 72.87 | 49.52 |
| $\mathcal{D}_{\mathrm{KL}}$ | $\mathcal{D}_{\lambda\text{-PR}}$ | 0.50 | 5.94 | 77.98 | 71.42 | 54.12 | 71.28 | 41.70 |
| $\mathcal{D}_{\mathrm{KL}}$ | $\mathcal{D}_{\lambda\text{-PR}}$ | 1.00 | 8.21 | 79.92 | 70.02 | 62.33 | 62.58 | 42.89 |
| $\mathcal{D}_{\mathrm{KL}}$ | $\mathcal{D}_{\lambda\text{-PR}}$ | 2.00 | 9.40 | 79.89 | 67.15 | 64.74 | 72.78 | 41.41 |
| $\mathcal{D}_{\mathrm{KL}}$ | $\mathcal{D}_{\lambda\text{-PR}}$ | 5.00 | 14.01 | 82.08 | 66.13 | 83.25 | 73.12 | 33.80 |
| $\mathcal{D}_{\mathrm{KL}}$ | $\mathcal{D}_{\lambda\text{-PR}}$ | 10.00 | 27.61 | **83.20** | 65.09 | 79.37 | 70.74 | 27.05 |

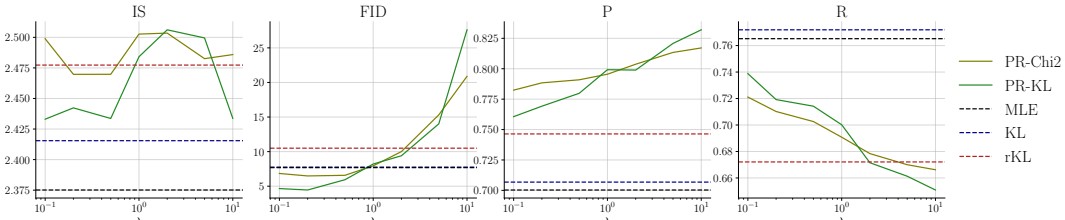

Figure D.11: MNIST: Glow models [27] are trained for different $\lambda$. From left to right, we plot IS ($\uparrow$), FID ($\downarrow$), P ($\uparrow$) and R ($\uparrow$). IS and FID are calculated using 50k samples, and P and R are calculated using 5k samples with $k = 3$ using Kynkäänniemi et al. [28]'s method. For comparison, we also report models trained with MLE (in black), for $\mathcal{D}_{\mathrm{KL}}$ (in blue) and for $\mathcal{D}_{\mathrm{rKL}}$ in red.

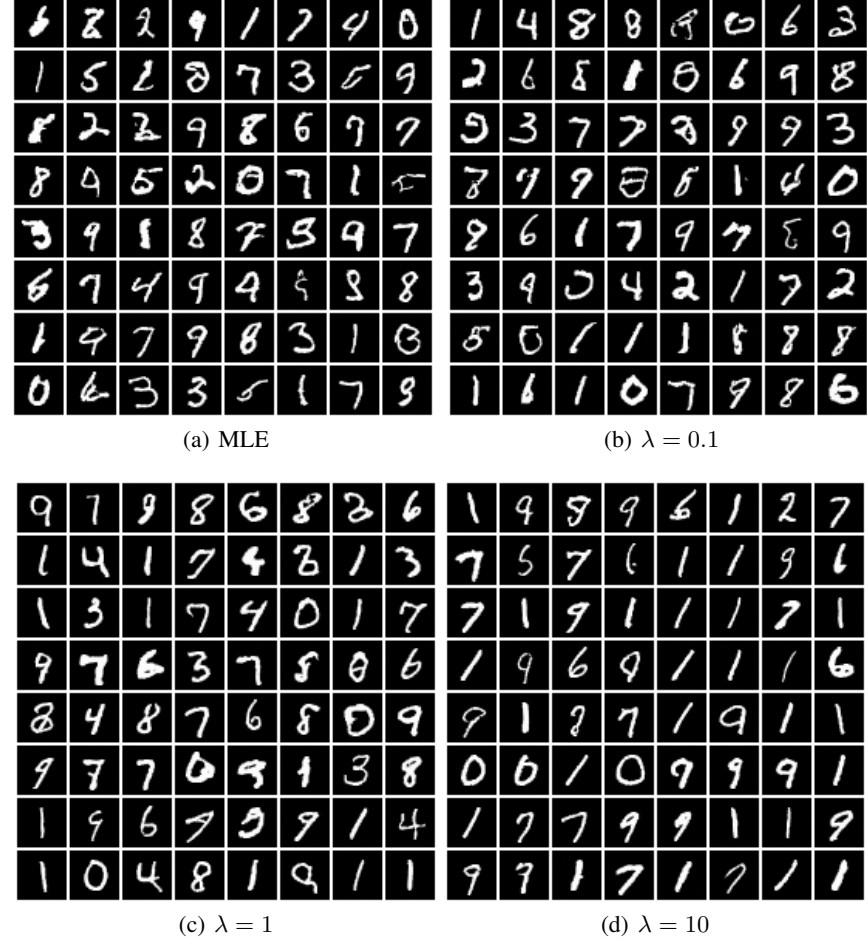

(a) MLE

(b) $\lambda = 0.1$

(c) $\lambda = 1$

(d) $\lambda = 10$

Figure D.12: MNIST: Samples are drawn from different models (only models trained with $g = f_{\chi^2}$). From 12(a) to 12(d), we observe that visual quality improves while diversity decreases. Models geared towards high recall, specifically MLE and those with $\lambda = 0.1$, are found to generate a wide variety of samples. However, these are characterized by lower precision, and approximately 20% of the generated samples are incoherent. On the contrary, the model trained with $\lambda = 10$ appears to focus on a smaller subset of modes, demonstrating higher precision but limited diversity. Specifically, this model primarily generates classes 1, 6, 7, and 9.

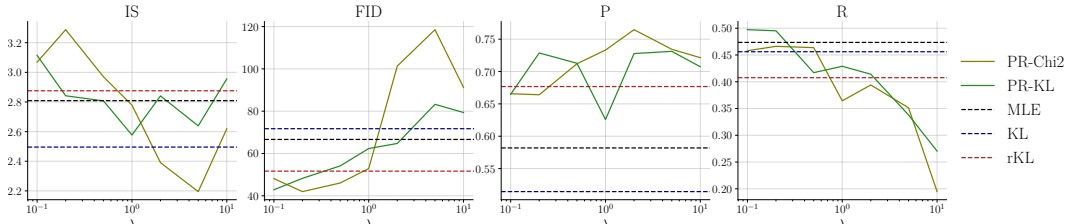

Figure D.13: FashionMNIST: Glow models [27] are trained for different $\lambda$. From left to right, we plot IS ($\uparrow$), FID ($\downarrow$), P ($\uparrow$) and R ($\uparrow$). IS and FID are calculated using 50k samples, and P and R are calculated using 10k samples with $k = 3$ using Kynkäänniemi et al. [28]'s method. For comparison, we also report models trained with MLE (in black), for $\mathcal{D}_{\mathrm{KL}}$ (in blue) and for $\mathcal{D}_{\mathrm{rKL}}$ in red.

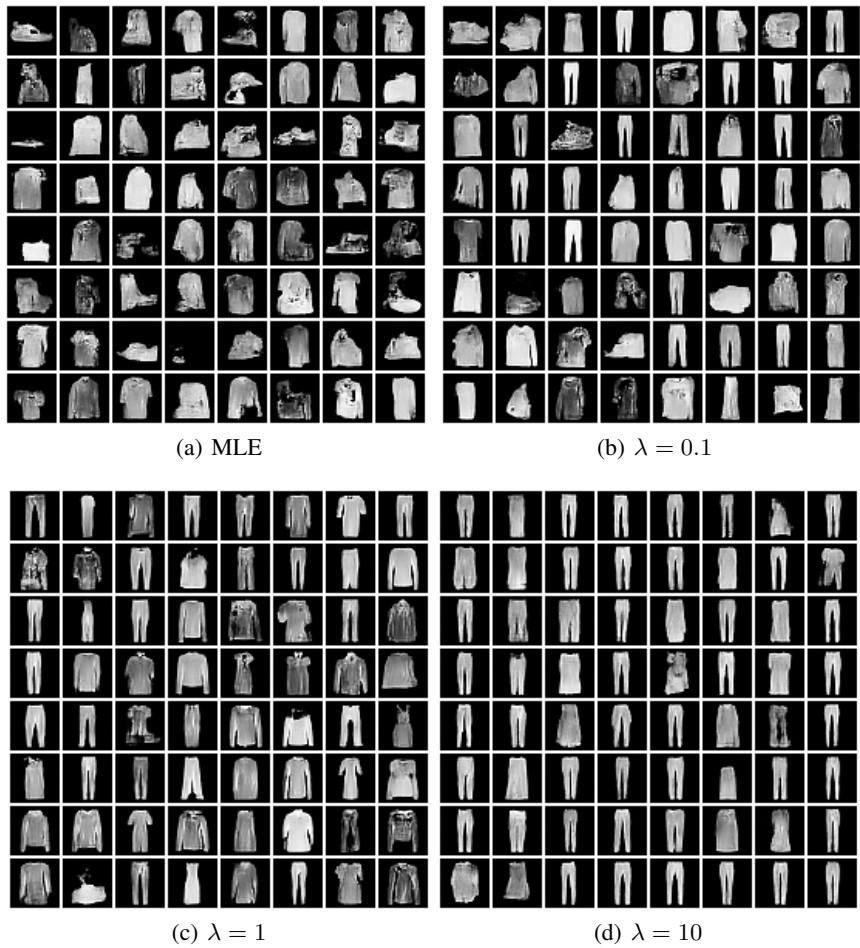

(a) MLE

(b) $\lambda = 0.1$

(c) $\lambda = 1$

(d) $\lambda = 10$

Figure D.14: FashionMNIST: Samples are drawn from different models (only models trained with $g = f_{\chi^2}$). From 14(a) to 14(d), we observe that visual quality improves while diversity decreases. Models geared towards high recall, specifically MLE and those with $\lambda = 0.1$, are found to generate a wide variety of samples. However, these are characterized by lower precision, and approximately 15% of the generated samples are incoherent. On the contrary, the model trained with $\lambda = 10$ appears to focus on a smaller subset of modes, demonstrating higher precision but limited diversity. Specifically, this model primarily generates the class "trouser".

### D.4 Training BigGAN on CIFAR-10 and CelebA64

In this section we give the details of the experiments on training BigGAN [7] models. To do this, we modify the official implementation of PyTorch[5] of BigGAN by Brock et al. [7]. $G$ and $T$ respectively count $4.3$M and $4.2$M parameters for CIFAR-10 and $32.0$M and $19.5$M for CelebA64. CIFAR-10's models are trained on 2 A100 80GB GPUs with a batch size of 128 for approximately 100k iterations ($\sim$ 7 hours), while CelebA64's models have been trained on 4 A1200 32GB GPUs with a batch size of 128 for 95k iteration ($\sim$ 20 hours). Quantitative results are reported in Table 2. The graphic representation of the results is in Figures D.16 and D.20. The training curves representative of FID, IS, P and R during training are reported in Figures D.17 and D.18.

---

[5]https://github.com/ajbrock/BigGAN-PyTorch

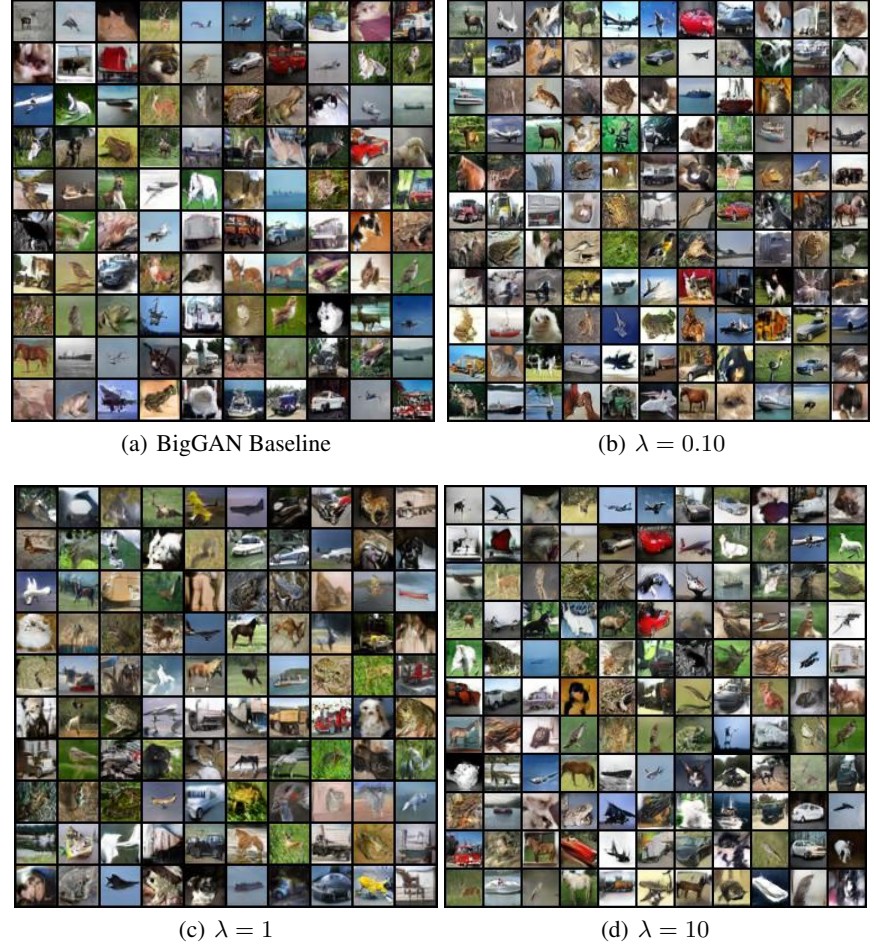

(a) BigGAN Baseline

(b) $\lambda = 0.10$

(c) $\lambda = 1$

(d) $\lambda = 10$

Figure D.15: CIFAR-10: Samples are drawn from different BigGAN trained on the PR-Divergence. We observe that when $\lambda$ increases, the recall decreases going to a various range of colored images to mostly brown images.

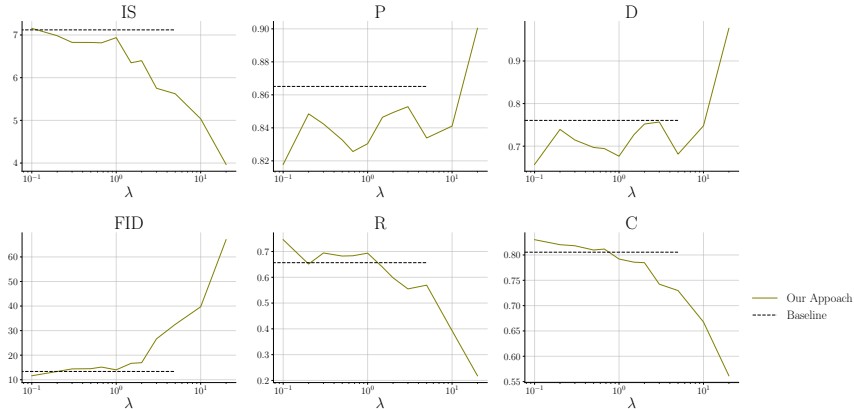

Figure D.16: CIFAR-10: BigGAN models [7] are trained for different $\lambda$. From left to right, we plot IS ($\uparrow$), P ($\uparrow$), D ($\uparrow$), FID ($\downarrow$), R ($\uparrow$) and C ($\uparrow$). IS and FID are calculated using 50k samples, and P and R are calculated using 10k samples with $k = 5$ using Kynkäänniemi et al. [28]'s method. For comparison, we also report a model trained hinge loss (in black), following the vanilla training procedure.

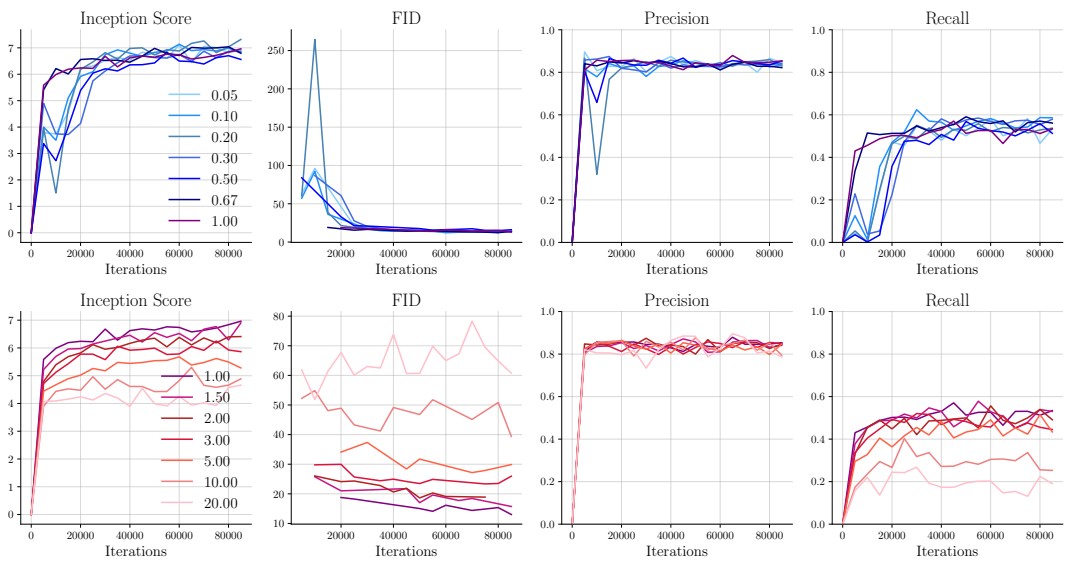

Figure D.17: CIFAR-10: Evolution of the IS, FID, P and R during training. We can observe that the precision quickly achieves its maximal value and saturates. We can also observe that the model is evicting modes of the target distribution early in the training, R is quickly saturates to different levels depending on $\lambda$. However, for low values of $\lambda$, the recall does not increase when $\lambda$ decreases, indicating that the model capacity limits the maximum recall.

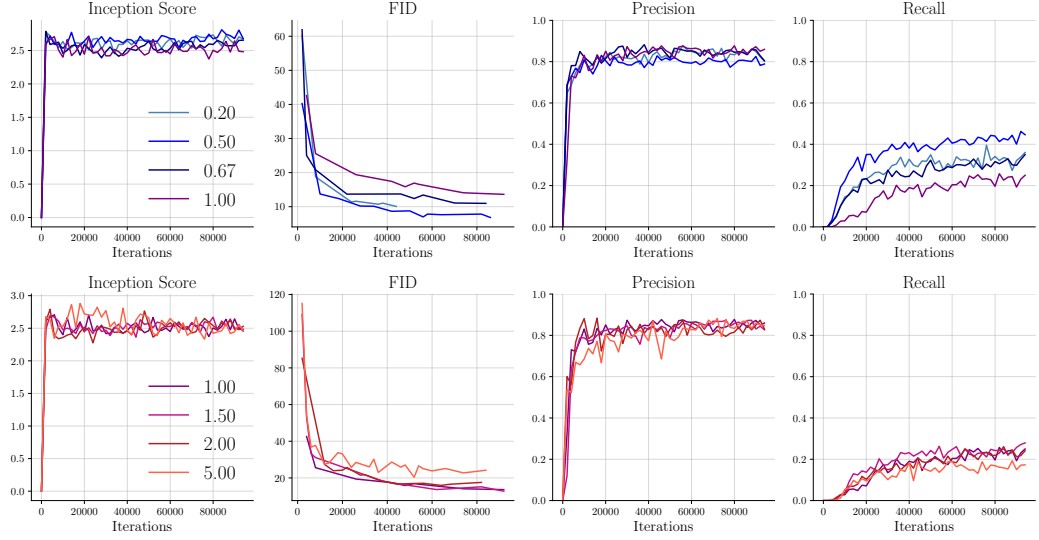

Figure D.18: CelebA Evolution of the IS, FID, P and R during training. We can observe that precision quickly achieves its maximal value and saturates. We can also observe that the model is evicting modes of the target distribution early in the training, R is quickly saturates to different level depending on $\lambda$.

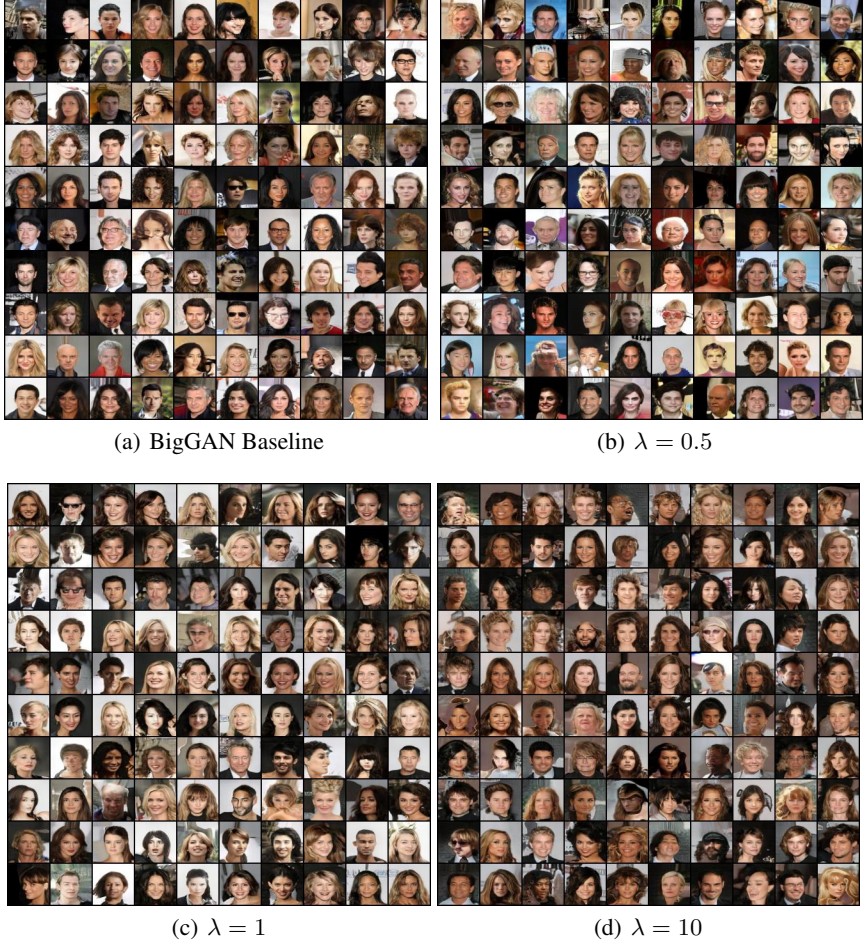

(a) BigGAN Baseline      (b) $\lambda = 0.5$

(c) $\lambda = 1$      (d) $\lambda = 10$

Figure D.19: CelebA64: Samples are drawn from different BigGANs with our apprach. For $\lambda = 0.5$, the model generated faces with various background, while the model generate only grey, black and brown backgrounds. However there are less artifacts on the latter model.

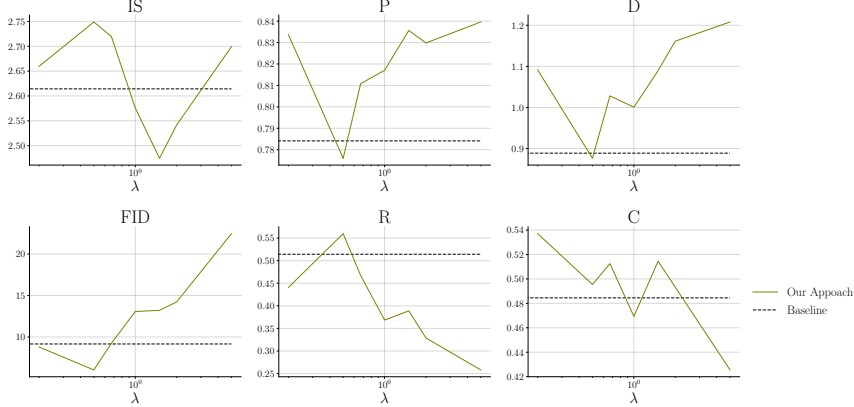

Figure D.20: CelabA64: BigGAN models [7] are trained for different $\lambda$. From left to right, we plot IS ($\uparrow$), P ($\uparrow$), D ($\uparrow$), FID ($\downarrow$), R ($\uparrow$) and C ($\uparrow$). IS and FID are computed using 50k samples and P and R are computed using 10k samples with $k = 5$ using Kynkäänniemi et al. [28]'s method. For comparison, we also report a model trained hinge loss (in black), following the vanilla training procedure.

## D.5 Fine-tuning BigGAN on ImageNet128 and FFHQ256.

In this section, we fine-tune the pre-trained BigGAN models. For ImageNet128, we use the weights of the model trained by Brock et al. [7], while for FFHQ256 we use a model pre-trained by our self, explaining why the FFHQ models is under performing. We train models on 4 V100 GPUs with a batch size of 128 for 10k iterations each. The graphic representation of the results is reported in Figures D.21 and D.22 and samples are shown in Figures D.5 and D.5.

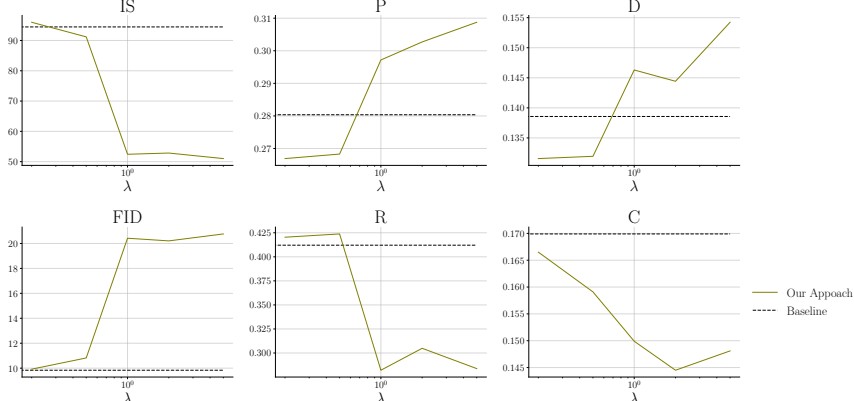

Figure D.21: ImageNet128: BigGAN models [7] are trained for different $\lambda$. From left to right, we plot IS ($\uparrow$), P ($\uparrow$), D ($\uparrow$), FID ($\downarrow$), R ($\uparrow$) and C ($\uparrow$). IS and FID are calculated using 50k samples, and P and R are calculated using 10k samples with $k = 5$ using Kynkäänniemi et al. [28]'s method. For comparison, we also report a model trained hinge loss (in black), following the vanilla training procedure.

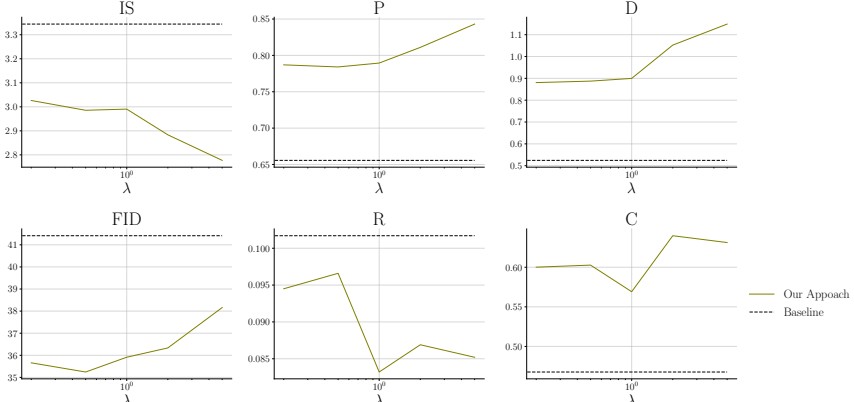

Figure D.22: FFHQ: BigGAN models [7] are trained for different $\lambda$. From left to right, we plot IS ($\uparrow$), P ($\uparrow$), D ($\uparrow$), FID ($\downarrow$), R ($\uparrow$) and C ($\uparrow$). IS and FID are computed using 50k samples and P and R are computed using 10k samples with $k = 5$ using Kynkäänniemi et al. [28]'s method. For comparison, we also report a model trained hinge loss (in black), following the vanilla training procedure.

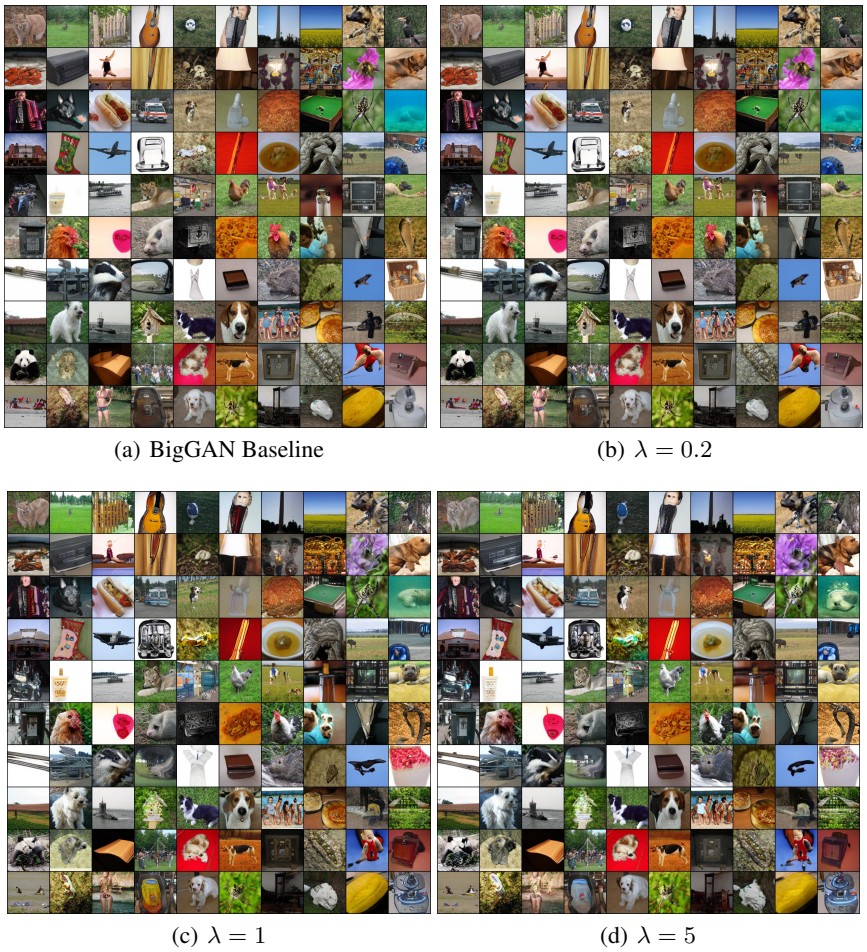

(a) BigGAN Baseline

(b) $\lambda = 0.2$

(c) $\lambda = 1$

(d) $\lambda = 5$

Figure D.23: ImageNet128: Samples are drawn from different BigGANs trained on the PR-Divergence.

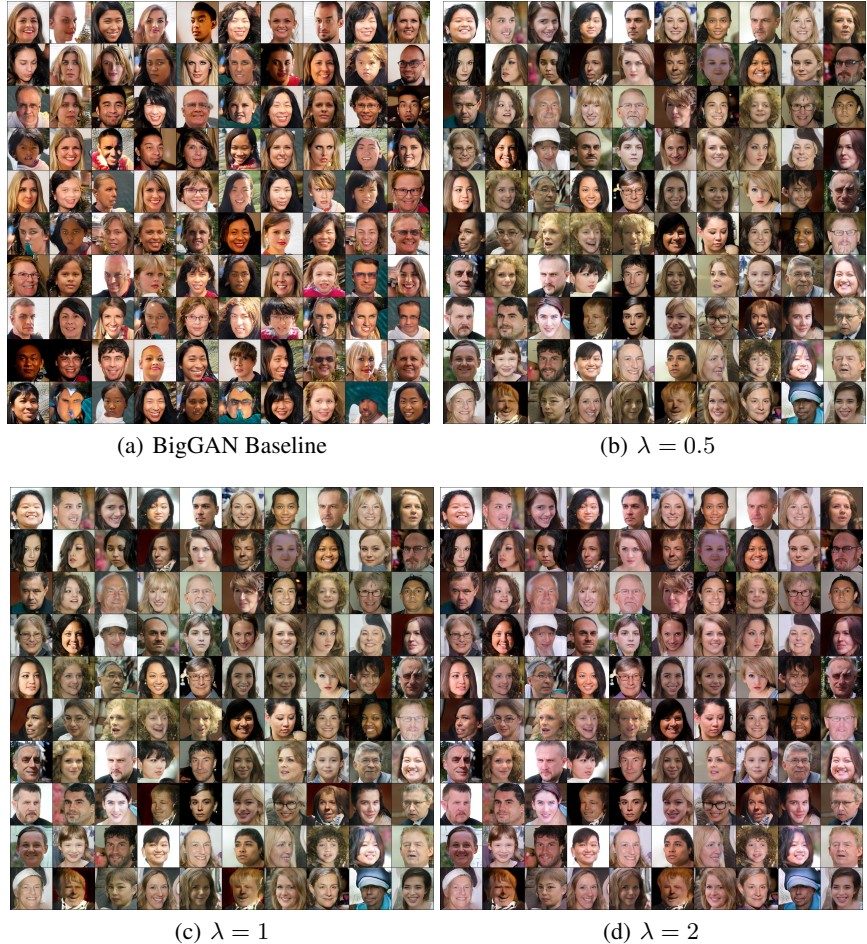

(a) BigGAN Baseline

(b) $\lambda = 0.5$

(c) $\lambda = 1$

(d) $\lambda = 2$

Figure D.24: FFHQ256: Samples are drawn from different BigGAN trained on the PR-Divergence.

# E    Training the AUC

In this paper, we propose a method to optimize any trade-off between precision and recall. In practice, we optimize a model to maximize any $\alpha_\lambda(P\|\widehat{P})$. We could also optimize for the area under the PR-Curves $\mathrm{PRD}(P, \widehat{P})(P, \widehat{P})$. In this section, we will show how we can compute the AUC in terms of $\alpha_\lambda(P\|\widehat{P})$ and train a model on a small dimension dataset and compare the results with the model trained on several $\lambda$. First, we must compute the AUC:

**Proposition E.1** (AUC under the $\partial\mathrm{PRD}(P, \widehat{P})$). *The area under the curve is:*

$$\mathrm{AUC} = \int_0^{+\infty} \frac{\alpha_\lambda(P\|\widehat{P})^2}{\lambda^2} \mathrm{d}\lambda \tag{51}$$

*Proof.* The AUC can be computed by integrating with respect to an angle $\theta$ in the first quadrant:

$$\mathrm{AUC} = \int_{\theta=0}^{\pi/2} \int_{u=0}^{r(\theta)} u\mathrm{d}u\mathrm{d}\theta = \int_0^{\pi/2} \frac{r^2(\theta)}{2} \mathrm{d}\theta. \tag{52}$$

Therefore, with $\lambda = \tan\theta$, we have $r(\theta) = \alpha_{\tan(\theta)}(P\|\widehat{P})/\sin\theta$ (see Figure E.25). Thus:

$$\text{AUC} = \int_0^{\pi/2} \frac{\alpha_{\tan(\theta)}(P\|\widehat{P})^2}{\sin^2\theta} d\theta \tag{53}$$

$$= \int_0^{\pi/2} \alpha_{\tan(\theta)}(P\|\widehat{P})^2 \left(1 + \frac{1}{\tan^2\theta}\right) d\theta \tag{54}$$

$$= \int_0^{+\infty} \alpha_\lambda(P\|\widehat{P})^2 \frac{\left(1 + \frac{1}{\lambda^2}\right)}{1 + \lambda^2} d\lambda \quad \text{with} \quad d\lambda = (1 + \tan^2\theta) d\theta, \tag{55}$$

$$= \int_0^{+\infty} \frac{\alpha_\lambda(P\|\widehat{P})^2}{\lambda^2} d\lambda. \tag{56}$$

$\square$

Using such expression, we can estimate the AUC and train model to directly optimize the AUC. It loses the focus on a specific trade-off between precision and recall, but instead optimizes overall performance. We trained a RealNVP model on a 2D 8 Gaussians example, similar to Section 6. We observe in Figure 27(d) that while the model trained to minimize the AUC performs poorly in terms of $\lambda = 0.1$ and $\lambda = 1$. However, we can clearly see that the AUC is greater and the model performs better in $\lambda = 1$ than the model trained for $\lambda = 1$ itself. We can also observe in Figure 27(d), the resulting model.

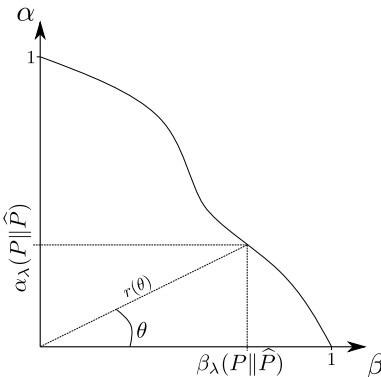

Figure E.25: Illustration of the change of variable to compute the AUC. Instead of parametrising the frontier $\partial\text{PRD}(P, \widehat{P})$ with $\lambda \in \mathbb{R} \cup [0, \infty]$, we take $\theta \in \left[0, \frac{\pi}{2}\right]$ with $\lambda = \tan\theta$.

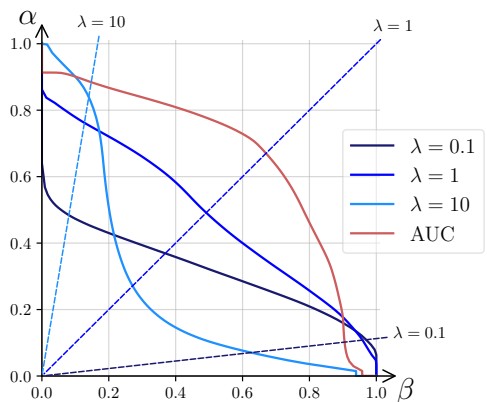

Figure E.26: PR-Curves corresponding to the models represented in Figure E.

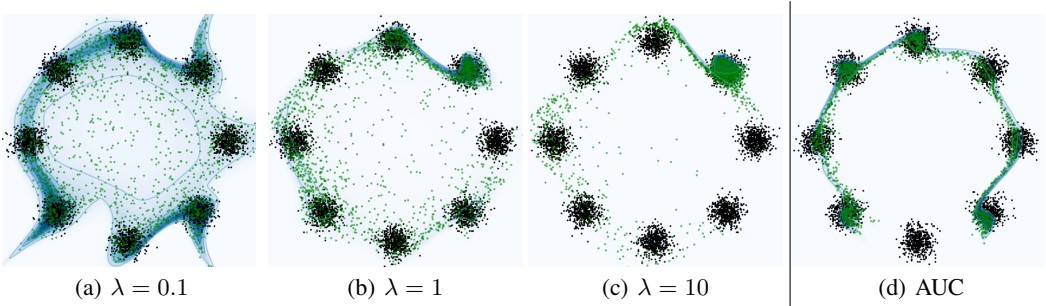

(a) $\lambda = 0.1$      (b) $\lambda = 1$      (c) $\lambda = 10$      (d) AUC

