# OpenReview forum: "Precision-Recall Divergence Optimization for Generative Modeling with GANs and Normalizing Flows"
_NeurIPS.cc/2023/Conference — NeurIPS 2023 poster_

### Official Review · Reviewer_VDUJ · 2023-06-15

**Soundness:** 3 good
**Presentation:** 3 good
**Contribution:** 2 fair
**Rating:** 6
**Confidence:** 4

**Summary:**

The paper proposes a way to train generative models such that they obtain a user defined tradeoff between sample fidelity and variety. Main findings are that the definition of precision and recall from previous works (by Simon et al.) can be written as a $f$-divergence (named PR-divergence) and that other $f$-divergences, such as KL or reverse-KL, can be formulated in terms of PR-divergence.

**Strengths:**

The strengths of the paper are thorough analysis of the $f$-divergences and solid theoretic foundation of PR-divergence. The paper provides interesting insights into what PR-tradeoff often used divergences, such as the KL, optimize for. Additionally, it proposes an algorithm to train generative models using an auxiliary divergence, since training models with $f$-GAN is difficult in practice.

**Weaknesses:**

The main weakness of the paper are empirical results. Although the method works as expected from the theory and it can be used to explicitly control the tradeoff between precision and recall it is far behind the current state-of-the-art.

**Questions:**

1. To further demonstrate the usefulness of explicitly controlling PR-tradeoff, I would like to see FID and PR curves from a following comparison: truncating BigGAN vs. applying classifier-free guidance to ADM models vs. training a separate model using different $\lambda$ to control the PR-tradeoff (on CIFAR-10, ImageNet 128x128 datasets).

2. Why do the FID results using BigGAN differ from the official results, reported by Brock et al. [1], on CIFAR-10 (13.37 vs. 14.73) and ImageNet 128x128 (9.83 vs. 8.7)? Two things that come to mind could be that you either use a validation set to compute FID or a different instance of Inception-V3 network (see App. A from [2] for other possible explanations). Also a minor note, on ImageNet 128x128 the baseline actually has the best FID instead of the model with $\lambda=0.2$.

Minor notes from checking the proofs:

3. There are different versions of defining $\lambda \in [0, \inf]$ in Definition 4.1 and Theorem 4.3. What is the difference?
4. Proof of Theorem in App. Eq. (6): Is the last term missing $\hat{p}(x)$, but it is correctly there in Eq. (7)?
5. App. Eq. (8): the first integral term has extra $($ and is missing $\textrm{d}\boldsymbol{x}$?
6. App. B.5: Title should be Proof of Theorem 4.4.
7. App. Eq. (30): Third term is missing $u_{\textrm{min}}$ from the denominator but it reappears correctly in Eq.(31)? Why does $c(1/u_{\textrm{max}})$ change suddenly to $c(0)$ (it cancels out but I’m still curious)?
8. Why in App. Eq (32) $\textrm{lim}$-term appears when differentiating w.r.t $u$?
9. Is there a typo in Theorem 5.2? Should it be $r(x) = \nabla g^*(T(x))$ instead of $f^*$?

[1]: Brock et al., Large Scale GAN Training for High Fidelity Natural Image Synthesis

[2]: Kynkäänniemi et al., The Role of ImageNet Classes in Fréchet Inception Distance


**Limitations:**

The authors adequately addressed the limitations of their work.

---

> ### Author Rebuttal · Authors · 2023-08-09
>
> Your comprehensive review and insightful comments are greatly valued, and we thank you for them. We would like to address the concerns and questions you raised:
>
> **1. Empirical Results and PR-tradeoff:**
> We acknowledge the importance of empirical results. We've added results for truncation on the Baseline BigGAN in the General Rebuttal. As observed, truncation primarily enhances the baseline precision, but it does not significantly improve recall.
>
> **2. FID Differences:**
> The discrepancy in FID for CIFAR-10 arises from our use of PyTorch, for which pretrained weights for BigGAN are unavailable. Consequently, we utilized a version of BigGAN that we trained ourselves. Additionally, as you rightly pointed out, we employed the PyTorch version of the Inception model, leading to FID differences from the official TensorFlow metrics. To address this, we recalculated FIDs for every different model using a consistent method.
>
> **3.  Addressing the Main Weakness:**
> The primary objective of our paper is to introduce a loss function that can effectively trade off between precision and recall. This loss can be applied to a wide range of generative models, including StyleGANXL and diffusion models. We firmly believe that our experimental setup validates the efficacy of our method. It's essential to note that our focus isn't on achieving only state-of-the-art results. Instead, we aim to showcase the versatility of our loss function. The notion of "state-of-the-art" is contingent on the specific model used and the user's preference for either recall or precision.
>
> **Typos and Clarifications:**
>
> - **Lambda Domain:** The domain of $ \lambda $ only differs in notation. In both cases, it encompasses all positive values, including 0 and $ +\infty $. We will ensure consistent notation in the final manuscript.
>
> - **Equation Typos:** We appreciate your keen observation of the typos in Eq (6), Eq (8), and the title for App B.5. These will be corrected.
>
> - **Bounds on Lambda:** We transitioned from 0 and $+\infty $ bounds on $ \lambda $ to $ 1/u_{max} $ and $ 1/u_{min} $ in theorem 4.4. The discrepancies in the proofs that you highlighted will be addressed. Thank you for pointing them out.
>
> - **Theorem 5.2 Notation:** The notation in theorem 5.2 is accurate. However, we acknowledge the reversed $f $ and $ g $ notations in the appendix. This will be rectified in the final manuscript.

---

> > ### Comment · Reviewer_VDUJ · 2023-08-15
> >
> > Thank you for the thorough answers to my feedback and the new interesting experiments!
> >
> > 1. Thank you for the additional experiment of explicitly controlling P&R tradeoff vs. truncation. Truncation is a way to trade variation to fidelity, thus, the result I hoped to see is that much higher Recall/Coverage could be achieved by explicitly controlling P&R tradeoff with the proposed method and I am glad to see the new data supporting this. ImageNet results are not inline with [1] and original results of Brock et al. [2] as truncation should improve fidelity at the cost of variation, what could be the reason for this discrepancy? As a minor note, to improve the quality of presentation of these results, I recommend showing a figure of P&R curves as in Fig. 6 from [1] to more easily observe the overall trends.
> >
> > 2. Thank you for checking FID calculation and making it consistent with the literature.
> >
> > 3. I agree that the experimental setup demonstrates that your method works as expected from the theory, and that “the best” generative model heavily depends on the downstream task that it is used for. However, it might be valuable for future work to point this direction of applying your method to diffusion models or larger scale generative models in the conclusions section.
> >
> > With the new data that you provided, I am happy to update my score.
> >
> > References:
> >
> > [1]: Kynkäänniemi et al., Improved Precision and Recall Metric for Assessing Generative Models
> >
> > [2]: Brock et al., Large Scale GAN Training for High Fidelity Natural Image Synthesis

---

### Official Review · Reviewer_Pocn · 2023-07-05

**Soundness:** 2 fair
**Presentation:** 3 good
**Contribution:** 2 fair
**Rating:** 3
**Confidence:** 4

**Summary:**

The paper introduces a technique for training generative models using an objective which approximates so called PR-divergence.  By adjusting a parameter, the paper claims that it is possible to train an array of models, from those that prioritize  high precision to those that prioritize high recall (mode seeking vs mode covering), as well as more balanced models. The supporting experiments involving BigGAN training on CIFAR10, CelebA64 and finetuning on ImageNET128 and FFHQ256 are described.

**Strengths:**

1. Authors consider the family of f-divergences and show that , given a trade-off parameter $\lambda$, for  a particular element
${\cal D}_{\lambda -PR}$  of this family, the minimization of  this element is equivalent to the maximization
of the value at $\lambda$ of the first component of so called PR curve from Saijidi et al [40].

2. As the minimization of ${\cal{D}}_{\lambda -PR}$  via f-GAN approach would fail, the authors describe a way to minimize a certain approximation to this objective and give an estimate for the error of this approximation.

3. The provided experiments show that  for NFs-GLOW on 2D synthetic dataset, MNIST and FMNIST, and for BigGAN on CIFAR-10, CelebA64,  the training with the approximation  objective and small $\lambda$ leads indeed to models with better mode covering, and, to a much lesser extent for BigGAN, the bigger tradeoff parameter $\lambda$ leads to better quality of generated samples.

**Weaknesses:**

Although the approach looks interesting, the paper raises several red flags that prevent me from endorsing it.

1. The experimental validation is limited, especially the gain in precision, i.e. the sample quality, for high tradeoff is insubstantial, compared with other methods. One example of this is the quality of samples on Figure D.11.d : all 100 supposedly high quality samples, at the highest tradeoff $\lambda=20$, are clearly of bad quality.

2. Similarly, the NFs results on Fig 1 in the simple 2D synthetic setup of eight gaussians do not really demonstrate excellent precision despite choosing the parameter lambda supposedly largely favoring the quality of generated samples.  Compare this with e.g. Wasserstein-GP GAN from 2017 paper (https://github.com/caogang/wgan-gp).

3. The code for the reproducibility check is not provided, despite the mentioning of an anonymized repository on page 8, the link is absent. This is especially disappointing as some evaluation numbers looks somewhat strange, like the P-value at $\lambda=20$ in Table 2 compared with FID, but it is not possible to try to reproduce this number or to see the details of its calculation.

4. Despite the fact that there are numerous variants in the literature of precision-recall scores, only one variant, based on k=3-NN only, from [27] is provided for evaluation of the models, which is known to not always work properly especially in regions of low density.

5. The related works description has some lacunas like "Reliable Fidelity and Diversity Metrics for Generative Models" (ICML'2020) from which another variant of PR scores can be used for evaluation, or "A Domain Agnostic Measure for Monitoring and
Evaluating GANs" (NeurIPS'2019).

6. Out of big diversity of GAN models, only the training of BigGan is tested with the proposed approximation objective.

7. In several places throughout the paper, the likelihood ratio $p(x)/\hat{p}(x)$ is used, e.g. in Theorem 5.2. However it is known that,  especially in high-dimensional spaces, the support of distributions is rather small and there can be large regions with  $\hat{p}(x)=0$ on which the methods involving the ratios as above give infinite or not well-defined answers. How to deal with this in each of the  encounters of this quantity ?


8. The clarity of the presentation can be improved, in particular the relation with other, more intuitive definitions of Precision and Recall scores from e.g. [27] and the "Reliable Fidelity and Diversity Metrics for Generative Models" (ICML'2020) paper,  is not explained.

9. The complexity of the training is not described.

10. The error bars are absent in the main experiment reported in Table 2.







**Questions:**

1. What is the complexity of the training procedure?

2. In several places throughout the paper the likelihood ratio $p(x)/\hat{p}(x)$ is used. How to deal with regions with $\hat{p}(x)=0$  in each of the  encounters of this ratio ?

3. In the experiments reported in Table 2, was only a single initialization used in each case to test the proposed method?

**Limitations:**

The limitations related with the complexity of the training procedure are not provided.  Also the limitations concerning the convergence of the proposed minimax procedure based on two different f are not discussed.

---

> ### Author Rebuttal · Authors · 2023-08-09
>
> Thank you for your thorough review and feedback on our paper. We appreciate the time and effort you've dedicated to understanding our work. We would like to address the concerns you raised:
>
> **1. Improving Recall:**
> While methods like rejection sampling, instance selection, or truncation primarily focus on enhancing precision, often at the expense of recall, our method aims to improve recall without significantly deteriorating precision. For the sake of completeness, we have added results for BigGAN using truncation for comparison to further illustrate this point.
>
> **2. Experimental Setup and Model Choice:**
> The primary objective of our experimental setup is to demonstrate the efficacy of our method in tuning any given model. Our theoretical framework establishes that our approach can be applied universally across GAN or NF architectures to balance precision and recall. We believe it's unnecessary to test our method on every existing architecture, especially considering the environmental impact of training large models like StyleGanXL. Moreover, in our 2D experiments, we intentionally used a model with limited expressivity to clearly illustrate how our method operates. Using a more complex model would have perfectly matched the distribution, distinguishing from real-world dataset and thus, making our method redundant.
>
> **3. Code Availability and Additional Results:**
> We apologize for the oversight regarding the code. We have provided the code to the area chair. Additionally, we've included extra results in the general rebuttal, showcasing the Precision Recall (Kynkäänniemi et al.) for k=5 and the Density Coverage (Naeem et al.) for k=5. These new metrics further validate our findings.
>
> Addressing the Questions:
>
> **Training Complexity:** As discussed in our paper, our training algorithm closely mirrors the original GAN training procedure. Consequently, both share similar algorithmic complexities, contingent on the neural network architecture used. We will make this clear in the paper, and add precise training times of our architectures.
>
> **Estimating PR Divergence:** We estimate the PR divergence using the primal form of the f-divergence. This primal form is estimated by sampling generated data points from $\widehat{ P}$:
>
> $$
> \mathcal{D}_{\lambda}(P \Vert\widehat{P}) = {\mathrm{E}} \_{\widehat{P}}
> \left[f \_{\lambda}\left(\frac{p(x)}{\widehat{p}(x)}\right)\right].
> $$
>
> Points where $ \widehat p(x) = 0$ will *not* be sampled for estimating the primal form. Thus, the value of the density ratio estimator on those points has no effect on the estimation of the PR divergence. We will make this clearer in the paper.
>
> **Metrics Evaluation:** Due to computational considerations, the metrics in Table 2 are evaluated on only one instance of the model. We observed similar numbers on other runs as well, and are open to adding error bars for the revised manuscript.
>
> **Related Works:**  The two papers you mention do make a good addition to the related works section - we thank you for pointing them out. However, we emphacize that our training method is focused on improving the precision and recall as defined in the works of [40, 43], and thereafter used in numerous follow-up works.
>
> **Complexity of training:** The complexity of our training method (both computational and memory) is comparable or no greater than that of existing approaches like f-GAN. We will make this clear in the paper.
>
> We hope that our clarifications address your concerns, and we are committed to refining our manuscript based on your valuable feedback.

---

> > ### Comment · Reviewer_Pocn · 2023-08-16
> > **Reply to Rebuttal**
> >
> > I'm thankful to authors for their response, which elucidated some points. However several issues were not properly addressed, among them:
> >
> > 1.From the paper's abstract: "our approach
> >  improves the performance of existing state-of-the-art models like BigGAN in terms
> > of either __precision__ or recall". The  weakness W1 was concerned with the __poor precision__ of the method applied to the BigGAN model. Indeed, the Figure  D.11.d  demonstrates the 100  samples obtained by the paper's method applied to BigGAN with the  tradeoff parameter set to $\lambda=20$, which corresponds to the proposed method's highest precision. The samples are obviously of unsatisfactory quality. Why the rebuttal answer is about improving recall? The question was about performance with respect to precision.
> >
> > In the rebuttal, in their response to the weakness W1 concerning the problematic precision, authors admitted that  "our method is aimed at improving recall", not  precision.  This changes the entire paper narrative concerning the "improving either precision or recall" and achieving "specified precision-recall trade-off" as described in the paper's abstract and throughout the paper. To the reviewer opinion  this important change of narrative requires an update of the paper and after that another round of reviewing.
> >
> > 2.The rebuttal contains the results of the  experiment on comparison of the proposed method with the truncation method applied to BigGAN model trained on ImageNet 128 dataset. The authors report very low  precision  values for the truncation method, in the range 20-28. The authors also state that the truncation method fails in this case as it diminishes both the recall and the precision. The results for the truncation method reported in the literature in this case are actually much higher, in the range 82-88, see eg "Improved precision and recall metric for assessing generative models" NeurIPS 2019, Figure 6. There is also the clear trend of the increasing  precision as the truncation diminishes on this Figure 6 from the literature. This raises the question as to whether there was a fair comparison between the methods, as the authors results on the baseline method do not match the results from the previous works.
> >
> > 3.The paper contains numerous encounters of the formulas with division by $\hat{p}(x)$ in the theoretical proofs of the principal results,  however $\hat{p}(x)=0$ in vast regions of the ambient space. When asked to clarify this, the authors didn't explain how to interpret rigorously these expressions in their theoretical proofs, but only mentioned how they approximate such quantity in their experimental part.
> >
> > 4.The provided at the rebuttal link to the anonymous github repository, which was last modified on the 8th of August, contains the code for one of the paper's experiments. However I could not use it for the thorough verification of the reproducibility of the paper's results  because of ethics consideration, as this would be unfair with respect to concurrent papers which didn't have an opportunity to have extra two months  for preparing their supportive materials.
> >
> > Because of the outlined issues I'm maintaining the score.

---

> > > ### Author Response · Authors · 2023-08-19
> > >
> > > **Response to Reviewer's Comments:**
> > >
> > > Thank you for your detailed feedback. We appreciate the time and effort you've put into reviewing our work. We'll address each of your concerns in turn:
> > >
> > > **Precision vs. Recall:** In our rebuttal, we emphasized the improvement in recall because our method is unique in its ability to enhance recall compared to other methods like truncation, reject, or instance selection. We understand the confusion arising from our rebuttal's narrative. There is no change in the paper's narrative: we propose a method to trade-off precision and recall, and for some dataset and some values of the parameters $\lambda$, we can achieve state of the art results. While we accept that the visual quality might not be properly assessed for datasets like CIFAR10, our method has demonstrated its effectiveness on MNIST, FashionMNIST, CelebA, FFHQ, and ImageNet.
> > >
> > > **BigGAN on ImageNet128:** In various experiments, particularly in "Improved Precision and Recall" [1], the authors utilized the TensorFlow pretrained version of BigGAN, likely the "BigGAN-deep" version, explaining their superior results (There is no mention to BigGAN in their official Github Repository). It's worth noting that the truncation experiments in [2] are not as straightforward as they might seem. Some works have shown that the relationship between truncation and precision and recall is not as clear-cut as expected (See Figure 6 in [2]). Moreover, the maximum precision and recall for k=5 and 10k samples (the setup we adopted based on your recommendations) for ImageNet128 are 84 and 82 respectively (See Table 5 in [2]). In [3], the precision and recall for BigGAN are reported as 86 and 35. This demonstrates that metrics from the literature can vary significantly and should be used to compare different models within the same framework as we did.
> > >
> > > **Mathematical Expressions of the likelihood ratio:** As we mention in the `Rebuttal`, the fact that $p$ and $\widehat p$" can be non-singular in practice does not pose any problems to the definition of PR divergences and the proofs. Wherever you find a likelihood ratio $p(x)/\widehat p(x)$ in the theorems and proofs, either in a function $f$, $\min$, or $\max$, it's multiplied by $\widehat p(x)$. Hence, for every $x$ in the sample space $\mathcal{X}$ where $\widehat p(x)=0$, we have $\widehat p(x) f(p(x)/\widehat p(x)) = 0$. In fact, for similar reasons, the framework of $f$-divergences can be extended to non-singular measures as outlined in [4].
> > >
> > > **Anonymous GitHub Repository:** We understand your ethical concerns regarding the verification of reproducibility using the provided code. Our intention was to provide as much support as possible for our claims, and we appreciate your understanding in this matter.
> > >
> > > Thank you again for your insights and constructive feedback. We hope this response addresses your concerns more comprehensively.
> > >
> > > [1]: Tuomas Kynkäänniemi, Tero Karras, Samuli Laine, Jaakko Lehtinen, and Timo Aila. Improved
> > > Precision and Recall Metric for Assessing Generative Models. In 33rd Conference on Neural
> > > Information Processing Systems (NeurIPS 2019), Vancouver, Canada., October 2019. arXiv:
> > > 1904.06991.
> > >
> > > [2]: Terrance DeVries, Michal Drozdzal, and Graham W. Taylor. Instance Selection for GANs,
> > > October 2020.  arXiv:2007.15255
> > >
> > > [3]: Axel Sauer, Katja Schwarz, and Andreas Geiger. StyleGAN-XL: Scaling StyleGAN to Large
> > > Diverse Datasets. In Special Interest Group on Computer Graphics and Interactive Techniques
> > > Conference Proceedings, pages 1–10, Vancouver BC Canada, August 2022. ACM. ISBN
> > > 978-1-4503-9337-9. doi: 10.1145/3528233.3530738
> > >
> > > [4]: Polyanskiy, Yury; Yihong, Wu (2022). Information Theory: From Coding to Learning (draft of October 20, 2022)

---

### Official Review · Reviewer_hgHq · 2023-07-06

**Soundness:** 4 excellent
**Presentation:** 3 good
**Contribution:** 4 excellent
**Rating:** 7
**Confidence:** 4

**Summary:**

The paper aims to balance the precision and recall of generative models including GANs and normalizing flows. Theoretically, the paper shows that the f-divergences to minimize can be reformulated as sums of PR-divergences, which provides a theoretical explanation between the optimization objective and the precision-recall trade-off. Empirically, the paper proposes a series of techniques to optimize the precision-recall divergence.

**Strengths:**

The paper is qualified to be accepted to NeurIPS in the following aspects:

1.	Novelty: Unlike previous heuristic works on the precision-recall trade-off, this paper established a fundamental theoretical understanding of the PR trade-off. To the best of our knowledge, the induced method is the first work to achieve a good PR trade-off from the perspective of divergence training.

2.	Significance: The proposed theoretical results can not only serve as a guide in training generative models but also as an extension of studies on the traditional PR trade-off, thus might be applicable to other tasks like retrieval and recommendation systems.

3.	Clarity: The paper is overall clear and well-written even with the heavy mathematics.

4.	Soundness: No obvious errors were found in the proofs. The theories are consistent with the empirical results (Fig. 4).


**Weaknesses:**

The main concern is the proposed optimization algorithm involves an additional hyperparameter $\lambda$, which needs to be tuned for a new dataset. Therefore, it might be an alternative to optimize the Area Under the PR Curve (AUPRC) instead of a single point in the PR curve, which has been studied in ranking problems [1,2,3,4]. The paper could be more instructive if the theoretical results can inspire the extension along this path.

Ref:

[1] Wang et. al. Momentum accelerates the convergence of stochastic auprc maximization. ICML, 2022.

[2] Wen et. al. Exploring the algorithm-dependent generalization of auprc optimization with list stability. NeurIPS, 2022.

[3] Cakir et. al. Deep metric learning to rank. CVPR, 2019.

[4] Chen et. al. Ap-loss for accurate one-stage object detection. T-PAMI, 2020.


**Questions:**

Please refer to the weaknesses.

**Limitations:**

Limitations are addressed.

---

> ### Author Rebuttal · Authors · 2023-08-09
>
> We are deeply appreciative of your thorough review and the positive feedback on our paper. Your insights are invaluable, and we would like to address the concern you raised:
>
> **Optimizing for AUC:**
> Your suggestion to optimize for the Area Under the Curve (AUC) is indeed insightful. In fact, we have previously explored this avenue for lower-dimensional datasets such as 2D, MNIST, and CIFAR-10. Our approach was formulated as maximizing
>
> $$
> \mathrm{AUC} = \int_{0}^{+\infty}\alpha_\lambda(P\Vert \widehat P)^2d\lambda.
> $$
>
> By parameterizing with  $\lambda=\tan(\theta)$ where $\theta\in[0, 1]$, we trained models to optimize the AUC. However, our observations indicated that the model behavior was closely aligned with $\lambda=1$. Given that this did not truly lead to  a trade-off between precision and recall, we chose not to include this study in the main paper. However, considering your interest, we are more than willing to add a dedicated section in the Appendix detailing our experiments with AUC optimization.

---

> > ### Comment · Reviewer_hgHq · 2023-08-13
> >
> > Thank you for your feedback. After reading other reviewers' comments and the authors' responses, I have decided to keep the original rating. Looking forward to your thoughts on optimizing the AUC.

---

### Official Review · Reviewer_oBuL · 2023-07-09

**Soundness:** 3 good
**Presentation:** 3 good
**Contribution:** 3 good
**Rating:** 6
**Confidence:** 4

**Summary:**

This paper focuses on the fine-grind optimization regarding the generation precision and recall. More specifically, considering the ambiguity of FID, the authors propose to directly optimize the precision and recall of generated images, by developing a certain class of f-divergence, that is, the PR-divergence. This paper also shows that a linear combination of the proposed PR-divergence can represent arbitrary existing f-divergence. This paper also proposes a practical primal-dual estimation on the PR-divergence, for relieving the gradient vanishing issues during training. The experimental results verify the feasibility of adjusting lambda to adjust the generation precision and recall.

**Strengths:**

This paper proposes a unique class of f-divergence named PR-divergence (up to an affine transform) that can explicitly optimize the precision and recall of GANs. The established theory of the PR-divergence is well aligned with the precision and recall of GANs, namely, the PR-curve. The author prove its connection to the vanilla f-divergence. The primal estimation of the divergence has also been provided for the ease of optimization, which is proved to be equivalent to minimizing the dual object in terms of the Bregman divergence. The presentation of this paper is clear for me. Experimental results have verified the effectiveness of optimizing different lambda-valued PR-divergence can alter the precision and recall.

**Weaknesses:**

One of my main concerns is regarding whether the PR-divergence can impede the generation quality, although it is proved to be able to alter the precision and recall during training. I was wondering, whether optimizing the PR-divergence for some lambda could at least retain the same best FIDs with other existing methods, under the same architecture and batch sizes. For example, as I read from the details in the supplementary material, the authors employed larger batch size (equal to 128) than the default setting (equal to 64) when training CIFAR-10 under the BigGAN architecture. However, the default setting of 64bs already achieved FIDs lower than 10, whereby the results are reported by https://github.com/POSTECH-CVLab/PyTorch-StudioGAN. All the FIDs for CIFAR-10 reported in this paper are larger than 10, even trained by 128bs. Why would this happen? Will the PR-divergence deteriorate the overall generation quality?

If I understood correctly, the proposed PR-divergence can be well applied to divergence-based or likelihood-based generative models. So I was wondering whether this method can be applied to diffusion models? Also for the normalization flows, did the authors compare the NFs on real-world datasets, other than on the synthetic datasets?

**Questions:**

1. The notation \leq or \geq when comparing two distributions should also be explained. For example, the notation \geq appears in  P\geq\beta\mu in Definition 3.

2. It seems Theorem 4.4 is not straightforward, where the proof or brief explanation is necessary.

3. The authors propose the g-divergence for the primal estimation. How the g-divergence is chosen in practice?

**Limitations:**

Please see my weakness.

---

> ### Author Rebuttal · Authors · 2023-08-09
>
> We sincerely appreciate the time and effort you dedicated to reviewing our paper, and we would like to address the concerns you raised:
>
> **1. Training Settings and FID Scores:**
> We acknowledge the discrepancies in the batch sizes used for training. Due to the unavailability of pretrained weights for BigGAN on CIFAR-10 (in PyTorch), we had to retrain the model ourselves. The batch size was chosen based on the computational resources available to us. Our primary goal is to demonstrate the tunability of models using our method, and we believe that a minor difference in FID scores (13.37 (ours) vs. 14.73 ([1])) does not significantly impact our main findings. It's worth noting that, as per your reference, most of the FID scores for CIFAR-10 are indeed higher than 10.
>
> **2. Applicability to Diffusion Models:**
> Our method can be extended to diffusion models. While the KL divergence in diffusion models is computed using a closed form between two Gaussians, our PR-Divergence doesn't have a closed form. However, we can approximate it using our technique. To achieve this, the discriminator would need to be $t$-dependent, as seen in some papers [2]. We are confident that this adaptation is feasible, but this would be an entirely different paper.
>
> **3. Training Normalizing Flows:**
> In our experiments (Section 6), we trained GLOW, a type of Normalizing Flow, on MNIST and Fashion MNIST using our method. Given the large model size of Normalizing Flows, we deemed it unnecessary to train on higher dimensions for the scope of this paper.
>
> **4. Choice of g-divergence and Theorem 4.4:**
>  In practice, we utilize the $\chi^2$ divergence. Theorem 4.4 essentially states that the auxiliary function $g$ should exhibit strong convexity. In Appendix C, we demonstrate that using an auxiliary $\chi^2$ divergence yields better results compared to KL. We will certainly provide a clearer explanation for the notation $P\geq \mu$ and offer more insights into Theorem 4.4 in our revised manuscript.
>
>
>
>
> [1]: Andrew Brock, Jeff Donahue, and Karen Simonyan. Large Scale GAN Training for High Fidelity Natural Image Synthesis, February 2019.
> arXiv:1809.11096 [cs, stat].
>
> [2]: Dongjun Kim, Yeongmin Kim, Wanmo Kang, and Il-Chul Moon. Refining generative process with discriminator guidance in score-based diffusion models. ArXiv, abs/2211.17091, 2022.

---

> > ### Comment · Reviewer_oBuL · 2023-08-18
> >
> > My concerns have been partially addressed the reviewers. However, I still feel unsatisfied on the response of experimental results. The authors claimed that there are no pre-trained weights. So they may need to train everything from scratch. However, why using bs=128 for CIFAR-10 (and even for CelebA) is still unclear for me, given that fact that most existing GANs use bs=64. I would expect bs=128 can further enjoy smaller FIDs, for which I didn't find in this paper. Btw, I found the proof of Theorem 4.4 by myself. The authors wrongly referred to Theorem 8 in the supplementary. After reading the comments from the other reviewers, I am still skeptical on the effectiveness of the PR divergence in terms of improving the overall generation quality. Given that this paper focuses the precision and recall, and does work, I tend to keep my score.

---

> > > ### Author Response · Authors · 2023-08-18
> > >
> > > **Response to Reviewer's Comments:**
> > >
> > > Thank you for your feedback and for taking the time to re-evaluate our work.
> > >
> > > Regarding the batch size concern:
> > > We understand the common practice of using `bs=64` for CIFAR-10 and CelebA in many GANs. In particular for older generation GANs such as SA-GAN, WP-GAN, or Progressive-GAN. Note that while larger batch sizes can sometimes lead to better FID scores, it is not a guaranteed outcome. The relationship between batch size and performance is complex and can be influenced by various factors, including model architecture, optimization techniques, and dataset specifics. In particular, the structure of GAN we have used in this work, *BigGAN*, is known for achieving better results for large batch size for low resolution ($32\times 32$ and $64\times 64$). As a matter of fact, more recent works such as  the BigGAN paper [1] and the StyleGAN-XL [2] are indeed advocating for larger batchsize:
> > > * "We begin by increasing the batch size for the baseline model, and immediately find tremendous benefits in doing so." [1]
> > > * "The main factors for BigGANs success are larger batch and model sizes." [2]
> > > * "We find it beneficial to use a large batch size [...] on lower resolution ($16^2$and $32^2$), similar to [1]." [2]
> > >
> > >
> > > Therefore, having trained multiple baseline models, and in accordance with  our computational ressources set up, we have find that the best model we could train is for a batch size of $128$.
> > >
> > > We apologize for the oversight in referencing Theorem 8 in the supplementary material. We appreciate your diligence in locating the proof of Theorem 4.4.
> > >
> > > Lastly, our primary focus in this paper is indeed on trading-off between precision and recall. We believe that our method offers a novel approach to balance these two aspects, and our experiments demonstrate its effectiveness in this regard. We'll continue our research to further validate and refine our approach.
> > >
> > > Thank you again for your insights and constructive feedback.
> > >
> > > [1] Andrew Brock, Jeff Donahue, and Karen Simonyan. Large Scale GAN Training for High Fi-
> > > delity Natural Image Synthesis, February 2019.
> > >
> > > [2] Axel Sauer, Katja Schwarz, and Andreas Geiger. StyleGAN-XL: Scaling StyleGAN to Large
> > > Diverse Datasets. In Special Interest Group on Computer Graphics and Interactive Techniques
> > > Conference Proceedings, pages 1–10, Vancouver BC Canada, August 2022. ACM. ISBN

---

> > > > ### Comment · Reviewer_oBuL · 2023-08-21
> > > >
> > > > Thanks for the update and clarification. Yes. As I have mentioned, I think this paper contributes to the way of controlling the precision and recall explicitly, and this is also the reason I would vote for positive scores. As from the authors response, they chose the best results by using bs=128, but still experienced similar FIDs with those of using bs=64. So I presumably thought the proposed metric did not improve the generation quality of GANs, since FIDs basically also evaluate the precision and recall in an implicit manner, by the Wasserstein distance between two Gaussians. After this rebuttal, I am keeping my original score.

---

### Official Review · Reviewer_jwf3 · 2023-07-21

**Soundness:** 3 good
**Presentation:** 4 excellent
**Contribution:** 3 good
**Rating:** 3
**Confidence:** 4

**Summary:**

This paper proposes a training method for generative models (normalizing flows and GANs), which can control the precision-recall trade-off of the generative models. The method is to design a new divergence named precision-recall divergence to bridge the precision-recall curve and the f-divergence. Then, the generative models optimized using the precision-recall divergence can control the precision-recall tradeoff by adjusting the hyperparameter lambda. The experiments show that the proposed method can control the precision-recall tradeoff and improve the performance of the baseline model (i.e., BigGAN).

**Strengths:**

1. This paper proposes a divergence that is directly related to the precision-recall curve, and the proposed method can control the precision-recall tradeoff during training.
2. The results show that the proposed method can control the precision-recall tradeoff.
3. The results show that the proposed method can improve the performance of BigGAN.

**Weaknesses:**

1. I think some combination of different kinds of f-divergence can also control the precision-recall tradeoff during the training. For example, KL-divergence + lambda * reverse KL-divergence. The paper does not compare with this kind of method, and does not explain the advantages of the proposed method in practice.
2. Although the existing methods including truncation and rejection sampling can not control the precision-recall during training, they can control the tradeoff through post-processing. The authors do not compare with these methods. What are the practical advantages of the proposed method compared to these methods?
3. The proposed method is not evaluated on larger resolutions (e.g., 512*512) and larger datasets. The ImageNet dataset is just used in the fine-tuning setting.
4. The proposed method is only evaluated on the BigGAN backbone. I am more interested in the use of StyleGAN, since StyleGAN is more widely used.
5. In Table 2, StyleGAN-XL performs much better than the proposed method.

**Questions:**

In addition to the theoretical one, what’s the practical advantages of the proposed method comparing to the existing methods such as truncation, rejection sampling, and combination of different f-divergences (e.g., KL + reverse KL)?

**Limitations:**

The authors have addressed the limitation in the paper.

---

> ### Author Rebuttal · Authors · 2023-08-09
>
> First, we would like to express our gratitude for the time and effort you dedicated to reviewing our paper. We appreciate the constructive comments. We would like to address the concerns you raised:
>
> **1. Combination of Different Kinds of $f$-divergence:**
> You rightly pointed out that a combination of different f-divergences, such as KL-divergence and reverse KL-divergence, might also control the precision-recall tradeoff. However, a combination of KL and reverse-KL, while being a suitable f-divergence, does not have a easy close form convex conjugate function $f^*$. The conjugate function depends on the Lambert function. So, to train a model to minimize a combination of KL and reverse KL, it would either require  two discriminators (one per divergence) or using a more complex version of our algorithm. While being a nice intuitive idea, combining different divergences is more complex than it appears to be. Secondly, even if we were to train for a combination of KL and reverse KL using multiple discriminators or another method, it is not a priori clear what PR trade-off is achieved by a combination, whereas the PR-divergence makes this trade-off very explicit. We discuss this in lines 193-199, but we will add a paragraph explaining this more clearly in the paper.
>
> **2. Comparison with Pre or Post processing Methods:**
> We recognize the importance of comparing our method with post-processing techniques like truncation and rejection sampling. Our method's primary advantage is the ability to control the precision-recall tradeoff during training. The other methods for sampling [1] or instance selection [2], can only shift the trade-off to improve the Precision while degrading the Recall. We have added in the general rebuttal some results for truncated latent distribution on the baseline model (BigGAN) and show that the method marginally improves the precision by limiting the recall.
>
> **3. Evaluation on Larger Resolutions, larger Datasets and more popular models:**
> We acknowledge the limitation regarding the evaluation of larger resolutions and datasets. Our primary focus is to demonstrate the efficacy and versatility of our method, and we chose the settings that best facilitate this.  Our choice of BigGAN was based on its compatibility with our method and the easier implementation/training procedure of BigGAN in practice. As a matter of fact, BigGAN is often used as a base to test new techniques and approaches [2, 3, 4]. We strongly believe that our experimental set up shows the efficiency of the method for a variety of models and datasets. In particular, a larger dataset, a higher dimension or larger model (like StyleganXL) would drastically increase the computational resources and environmental impact.
>
>
> **4. Practical Advantages Over Existing Methods:**
> Beyond the theoretical advantages, our method offers several practical benefits, the most important being:
>
> **Explicit Trade-off:** The precision-recall tradeoff is made explicit in our approach, allowing for more predictable and controlled outcomes during training. We directly optimize well-established precision and recall measures, which others do not.
>
> Also, our method provides:
>
> **Minimal Cost:** One of the standout features of our method is that it introduces minimal computational overhead. The complexity of training remains unchanged, ensuring that the benefits of our approach come at no additional cost.
>
> **Improved Recall:** Our method has the potential to enhance recall, a feat that other techniques like rejection sampling, importance sampling, and truncation cannot achieve.
>
>
> [1] Humayun, A. I.; Balestriero, R.; and Baraniuk, R. 2022. Polarity Sampling: Quality and Diversity Control of Pre-Trained Generative Networks via Singular Values. ArXiv:2203.01993 [cs].
>
> [2] Terrance DeVries, Michal Drozdzal, and Graham W. Taylor. Instance Selection for GANs,
> October 2020. arXiv:2007.15255
>
> [3] Hanxiao Liu, Andrew Brock, Karen Simonyan, and Quoc V Le. Evolving normalizationactivation layers. In NeurIPS, 2020.
>
> [4] Mario Lucic, Michael Tschannen, Marvin Ritter, Xiaohua Zhai, Olivier Bachem, and Sylvain Gelly. High-fidelity image generation with fewer labels. arXiv:1903.02271, 2020

---

> > ### Comment · Reviewer_jwf3 · 2023-08-21
> >
> > Thanks for the response. However, my major concerns are not fully addressed.
> > 1. Combination of Different Kinds of f-divergence.
> >
> > The authors stated that it needs to incorporate two discriminators. Actually, it is not needed, and two different divergences can be incorporated into the objective function directly. There are papers discussing this kind of method. Thus, I am still not sure about the advantages of the proposed method over the method of combination of different divergences.
> >
> > 2. Comparison with Pre or Post processing Methods, and the practical benefit of the proposed method
> >
> > I still don't get the point about the practical advantage of controlling the precision-recall tradeoff during training. For example, if we need a specific precision-recall tradeoff, for the proposed method, we need to train many different models and select the model that satisfies our requirement. But for the Post processing Methods, we just need to train one model and then adjust the sampling scheme. This is good for the environment (since the authors mentioned the environment in the response).
> >
> > 3. Evaluation on Larger Resolutions, larger Datasets and more popular models
> >
> > I still believe that generalizing the proposed method to a larger dataset and to StyleGAN is important.
> >
> > Besides, I agree with some points raised by Reviewer Pocn.
> >
> > Overall, I would like to change my score to reject.

---

### Author Response · Authors · 2023-08-08
**Link for Anonymized Github**

Dear Area Chairs,

Since links are not allowed in the Rebuttal Sections, we are required to submit any external links directly to you.
Reviewer Pocn is rightfully observing the absence of the anonymized Github link in the paper. To rectify this mistake, we would like to share the link with you for approval.

https://anonymous.4open.science/r/PrecisionRecallGan-004B/README.md

Best regards,

the Authors

---

### Author Rebuttal · Authors · 2023-08-09

Dear Reviewers,

In response to the feedback from the reviewers, we have added some additional results for a more comprehensive comparison. Two reviewers inquired about a comparison with other methods such as truncation, and one recommended adding Density and Coverage metrics from Naeem et al. Moreover, Precision and Recall are computed for $k=5$. Below are the results for four different datasets:



### CIFAR-10

| Model | FID | Precision | Recall | Density | Coverage |
|---|---|---|---|---|---|
| Baseline BigGAN $\psi=1.0$ | 13.38 | 86.54 | 65.63 | 0.76 | 0.81 |
| Baseline BigGAN $\psi=0.7$ | 22.25 | 90.81 | 48.01 | 0.90 | 0.67 |
| Baseline BigGAN $\psi=0.5$ | 36.10 | 92.34 | 22.11 | 1.00 | 0.48 |
|---|---|---|---|---|---|
| $\lambda=0.05$ | 13.88 | 85.29 | 68.40 | 0.72 | 0.83 |
| $\lambda=0.10$ | 11.62 | 81.78 | 74.58 | 0.66 | 0.83 |
| $\lambda=0.20$ | 13.36 | 84.85 | 65.13 | 0.74 | 0.82 |
| $\lambda=0.30$ | 14.41 | 84.24 | 69.42 | 0.71 | 0.82 |
| $\lambda=0.50$ | 14.50 | 83.27 | 68.23 | 0.70 | 0.81 |
| $\lambda=0.67$ | 15.15 | 82.57 | 68.34 | 0.69 | 0.81 |
| $\lambda=1.00$ | 15.26 | 81.96 | 72.51 | 0.65 | 0.80 |
| $\lambda=1.50$ | 16.68 | 84.64 | 63.92 | 0.73 | 0.79 |
| $\lambda=2.00$ | 18.25 | 79.53 | 72.90 | 0.59 | 0.78 |
| $\lambda=3.00$ | 26.66 | 85.28 | 55.49 | 0.76 | 0.74 |
| $\lambda=5.00$ | 32.54 | 83.39 | 56.94 | 0.68 | 0.73 |
| $\lambda=10.00$ | 39.69 | 84.11 | 39.29 | 0.75 | 0.67 |
| $\lambda=20.00$ | 67.03 | 89.64 | 20.52 | 0.97 | 0.56 |


### CelebA 64

| Model | FID | Precision | Recall | Density | Coverage |
|---|---|---|---|---|---|
| Baseline BigGAN $\psi=1.0$ | 9.17 | 78.48 | 51.36 | 0.89 | 0.49 |
| Baseline BigGAN $\psi=0.7$ | 23.72 | 87.82 | 31.11 | 1.29 | 0.49 |
| Baseline BigGAN $\psi=0.5$ | 43.64 | 91.01 | 11.54 | 1.53 | 0.39 |
|---|---|---|---|---|---|
| $\lambda=0.2$ | 8.79 | 83.37 | 44.07 | 1.09 | 0.54 |
| $\lambda=0.5$ | 6.03 | 77.60 | 55.98 | 0.88 | 0.50 |
| $\lambda=0.7$ | 9.24 | 81.08 | 46.71 | 1.03 | 0.51 |
| $\lambda=1.0$ | 13.07 | 81.70 | 36.85 | 1.00 | 0.47 |
| $\lambda=1.5$ | 13.21 | 83.56 | 38.89 | 1.09 | 0.51 |
| $\lambda=2.0$ | 14.23 | 82.98 | 32.87 | 1.16 | 0.49 |
| $\lambda=5.0$ | 22.44 | 84.04 | 25.67 | 1.21 | 0.43 |

### ImageNet 128


| Model | FID | Precision | Recall | Density | Coverage |
|---|---|---|---|---|---|
| Baseline BigGAN $\psi=1.0$ | 9.84 | 27.97 | 40.92 | 0.14 | 0.17 |
| Baseline BigGAN $\psi=0.7$ | 11.39 | 23.12 | 31.77 | 0.11 | 0.15 |
| Baseline BigGAN $\psi=0.5$ | 15.49 | 20.25 | 20.08 | 0.10 | 0.14 |
|---|---|---|---|---|---|
| $\lambda=0.2$ | 9.92 | 26.69 | 42.04 | 0.13 | 0.17 |
| $\lambda=0.5$ | 10.82 | 26.83 | 42.38 | 0.13 | 0.16 |
| $\lambda=1.0$ | 20.42 | 29.72 | 28.21 | 0.15 | 0.15 |
| $\lambda=2.0$ | 20.21 | 30.27 | 30.49 | 0.14 | 0.14 |
| $\lambda=5.0$ | 20.76 | 30.87 | 28.38 | 0.15 | 0.15 |

### FFHQ 256

| Model | FID | Precision | Recall | Density | Coverage |
|---|---|---|---|---|---|
| Baseline BigGAN $\psi=1.0$ | 41.42 | 65.54 | 10.02 | 0.52 | 0.47 |
| Baseline BigGAN $\psi=0.7$ | 56.44 | 76.60 | 4.83 | 0.70 | 0.41 |
| Baseline BigGAN $\psi=0.5$ | 82.04 | 84.51 | 1.50 | 0.89 | 0.32 |
|---|---|---|---|---|---|
| $\lambda=0.2$ | 35.66 | 78.70 | 9.45 | 0.88 | 0.60 |
| $\lambda=0.5$ | 35.24 | 78.41 | 9.66 | 0.89 | 0.60 |
| $\lambda=1.0$ | 35.91 | 78.95 | 8.32 | 0.90 | 0.57 |
| $\lambda=2.0$ | 36.33 | 81.10 | 8.69 | 1.05 | 0.64 |
| $\lambda=5.0$ | 38.16 | 84.31 | 8.52 | 1.15 | 0.63 |

**Observations on the Tables:**

1. **Cifar and CelebA:**
   - Truncation primarily enhances Precision at the expense of Recall, yielding results comparable to our method.
   - However, when it comes to improving Recall, our method stands out. By training models with $\lambda < 1$, we've been able to boost the baseline Recall, a feat truncation fails to achieve.

2. **Imagenet:**
   - Truncation appears to be counterproductive, diminishing both Precision and Recall. This suggests that the mean of the Gaussian might be mapped outside the support of the target distribution.

3. **FFHQ:**
   - Our method demonstrates a superior trade-off compared to truncation. For instance, at a Precision of 84%, our method achieves a Recall of 8.52% versus truncation's 1.5%.

---


Thank you for the constructive feedback. We've made efforts to address the concerns raised and hope that the additional results and explanations provided here shed more light on our approach.

---

### Decision · Program_Chairs · 2023-09-21

**Decision:**

Accept (poster)

**Comment:**

This paper proposes a family of divergences for training generative models (e.g., GANs) over a user-defined trade-off between precision and recall. They is some good theoretical analysis, e.g., w.r.t. f-divergences and empirical results support the claim that this approach can be used to improve precision / recall. The work is interesting, novel, and important / well motivated and justified, it appears that the presentation and while there were concerns about soundness of some of the theory, sufficient clarifications were made. It would be nice to see this method applied to more generative models (e.g, styleGAN, etc), but I believe BigGAN was sufficient for demonstrating the core ideas work. I therefore recommend this paper to be accepted as a poster.